# Multi species asymmetric simple exclusion process with impurity activated flips

Amit Kumar Chatterjee[1*], Hisao Hayakawa[1,2†],

**1** Yukawa Institute for Theoretical Physics, Kyoto University, Sakyo-ku, Kyoto 606-8502, Japan
**2** Center for Gravitational Physics and Quantum Information, Yukawa Institute for Theoretical Physics, Kyoto University Sakyo-ku, Kyoto 606-8502, Japan

\* ak.chatterjee@yukawa.kyoto-u.ac.jp, † hisao@yukawa.kyoto-u.ac.jp

November 28, 2022

## Abstract

We obtain an exact matrix product steady state for a class of multi species asymmetric simple exclusion process with impurities, under periodic boundary condition. Alongside the usual hopping dynamics, an additional flip dynamics is activated only in the presence of impurities. Although the microscopic dynamics renders the system to be non-ergodic, exact analytical results for observables are obtained in steady states for a specific class of initial configurations. Interesting physical features including negative differential mobility and transition of correlations from negative to positive with changing vacancy density, have been observed. We discuss plausible connections of this exactly solvable model with multi lane asymmetric simple exclusion processes as well as enzymatic chemical reactions.

# 1   Introduction

Non-equilibrium stochastic processes are ubiquitous in nature, with wide range of applicability in physics [1–3], chemistry [1], biology [4] and interdisciplinary areas [5–7]. In fact, even in one dimension, several models of non-equilibrium statistical mechanics exhibit surprisingly rich physical phenomena including phase transitions [8] along with the important feature of analytical tractability [9]. The asymmetric simple exclusion process (ASEP) [2, 10–12] is broadly regarded as a paradigmatic model for non-equilibrium transport processes as diverse as traffic and pedestrian flow [13], mRNA translation by ribosomes [14] and motor protein transport through single filaments [15, 16] etc.

Apart from its extensive success in modeling numerous real-world phenomena, ASEP and its variations have been instrumental in understanding the mathematical structures and physical characteristics of generic non-equilibrium steady states and dynamics [2, 12, 17–20]. In particular, the exact steady states of the totally asymmetric simple exclusion process (TASEP) [12, 21] and the general ASEP [22] with open boundary conditions, have been obtained using matrix product ansatz. Except for the infinite dimensional representations [21, 22], interesting and especially useful finite dimensional matrix representations have been achieved for corresponding quadratic algebra for certain conditions on the transition rates [23, 24]. The matrix

product ansatz has been extremely effective in deriving the non-equilibrium steady states of several generalizations of ASEP including two species [25] and multi-species processes [26], see Ref. [27] for a detailed review. In fact, the stationary state for the multi species TASEP has been solved remarkably by a different method of multiline queuing process [28], which is explored further in terms of combinatorial $R$ in crystal base theory [29]. Several two point and three point correlations have been studied analytically in the multi species TASEP [30] and in-homogeneous multi species TASEP with species dependent rates have been analyzed [31, 32]. The multi species ASEP has also been investigated with integrable open boundary conditions [33] and matrix product solutions are found [34]. Due to the connection of TASEP to integrable spin chains [35], the algebraic Bethe Ansatz has been applied to study the dynamics of TASEP [36] and ASEP [37] with open boundaries. Interestingly, ASEP belongs to Kardar-Parisi-Zhang universality class [38] with dynamic exponent $\frac{3}{2}$ [39, 40]. With the aid of Monte Carlo simulations and several improved versions of mean-field theories, TASEP has also been generalized to non-conserved dynamics [41], two-lane [42–44] and multi-lane [45–48] models relating to traffic flow and complex networks [49].

It is quite natural to expect the presence of multiple species of particles with a variety of microscopic dynamics in a system in general. Often due to the distinction between the dynamics of different species, some species are referred to as *impurities* and give rise to fascinating physical and mathematical structures. For example, the presence of a single impurity which hops with a different rate and allows overtaking of ordinary particles in the TASEP on a periodic lattice, leads to a matrix product steady state with six distinct phases including the creation of a shock in one of the phases [50]. This impurity model has been generalized by considering bidirectional asymmetric hopping of the ordinary particles [51] which allows for finite dimensional matrix representations in certain regions of the parameter space, in comparison to the infinite dimensional representation in Ref. [50]. A phase transition arising from the motion of the single impurity in the direction opposite to the ordinary particles has also been observed [52]. The long time limit behavior of the TASEP with a single impurity has been solved using the Bethe Ansatz [53] and the diffusion constant of the impurity has been calculated from both the Bethe Ansatz [53] and the matrix product ansatz [54]. Remarkably, a disordered ASEP with species dependent hop rates, has been shown to exhibit Bose-Einstein condensation [55]. A variation of the ASEP with ordinary particles and many impurities has been considered in Ref. [56], where the impurities are not allowed to hop to the vacant neighbors but they can exchange positions with ordinary particles. Interestingly, such an impurity model in [56] possesses a different scaling exponent $\frac{5}{2}$ in comparison to the usual KPZ exponent $\frac{3}{2}$ for the ASEP without impurities [39, 40]. Other than the disorders or impurities associated with the particles themselves, there are many exciting studies with position dependent or site-wise disorders for ASEP [57–63].

In this article, we study a class of the multi-species ($I = 1, 2, \ldots, \mu$) ASEP in the presence of impurities, under periodic boundary conditions. In addition to the usual hopping of particles to vacant sites in ASEP, we consider flips of different species among each other (e.g. species $I$ transforming to species $J$ and vice versa). Importantly, these flip processes are initiated *only in the presence of a special type of particles* (as nearest neighbors) that we denote *impurities*. These impurities activate the flip processes. Therefore, we name this non-equilibrium stochastic process to be *multi species asymmetric simple exclusion process with impurity activated flips* ($\mu$-ASEP-IAF). Note that the total number of impurities, along with that of the vacancies, remain conserved in the $\mu$-ASEP-IAF. Specifically, we emphasize that the flip processes between two non-conserved species ($I, J$) do not occur through

the interaction with any non-conserved species $K$ $(= 1, 2, \ldots, \mu)$ at the nearest neighbors. Thus, the microscopic dynamics considered here is different from previously studied models like TASEP with internal degrees of freedom [64] and multi-species reaction-diffusion processes [65, 66]. Notably, the distinction in the microscopic dynamics also makes $\mu$-ASEP-IAF *non-ergodic* in nature in contrast to the ergodic models [64–66]. We should mention that the non-ergodicity of exactly solvable models is related to undecidability of thermalization in integrable models [67].

The motivations for studying the $\mu$-ASEP-IAF are as follows. (i) We aim to obtain an exact non-equilibrium steady state of the $\mu$-ASEP-IAF under periodic boundary condition, so that it would be an important addition to the category of exact solvable models in disordered systems. (ii) The $\mu$-ASEP-IAF being non-ergodic, it would be interesting to derive exact analytical expressions for partition function and observables for suitable choice of initial configurations and compare the corresponding steady state results with that of a random initial configuration. (iii) The $\mu$-TASEP-IAF can be mapped to multi-lane TASEP which is a basic model for multi-lane traffic flow. Different species of particles in $\mu$-ASEP-IAF play the roles of particles in different lanes of multi-lane TASEP and the impurities in $\mu$-ASEP-IAF act as bridges between lanes that allow particles to exchange lanes in multi lane TASEP. See Appendix A for details. (iv) Considering the conserved impurities as enzymes ($E$) and different non-conserved species as substrates ($S$) and products ($P$), the flip process of $\mu$-ASEP-IAF can be thought as an enzymatic chemical reaction like $S + E \rightarrow P + E$, which is a crude approximation of the Michaelis-Menten reaction scheme $S + E \rightleftharpoons SE \rightarrow P + E$ [68–70]. See Appendix C for details. Notably, both the mappings in (iii) and (iv) would not be possible if the impurities could also flip to other species.

Below we briefly summarize our main results.
(i) We find that the steady states of $\mu$-TASEP-IAF (totally asymmetric hopping) and $\mu$-ASEP-IAF (bidirectional hopping) under periodic boundary conditions, can be obtained exactly as matrix product states, where distinct matrices represent different components (each species, impurity, vacancy) of the system. We provide explicit finite dimensional matrix representations for the totally asymmetric case, whereas the matrices for the general asymmetric case are found to be infinite dimensional .
(ii) For a specific choice of initial configuration, we could analytically calculate the partition function in the sector of allowed configurations in the steady state and consequently the observables of interests (average densities of non-conserved species, currents and spatial correlations). The analytical results are in agreement with Monte Carlo simulations. For a fixed set of input parameters, we show considerable quantitative deviations between steady state observable values for different initial configurations, establishing the initial configuration dependence or non-ergodicity of the dynamics.
(iii) Two-point nearest neighbor correlations exhibit interesting non-trivial behaviors. Particularly, with the variation of the vacancy density, we observe characteristics like certain correlations changing signs i.e. varying from negative to positive with some intermediate zero correlation point, and, non-monotonic behavior with both local maximum and local minimum.
(iv) We find *negative differential mobility* in $\mu$-ASEP-IAF. For special choices of hopping rates, both the drift current and flip current decrease with increasing bias giving rise to negative differential mobility.

The article is organized as follows. In Sec. 2 we describe the $\mu$-TASEP-IAF in details and show that the steady state can be achieved using matrix product ansatz. The analytical calculation of partition function starting from a suitably chosen initial configuration is presented

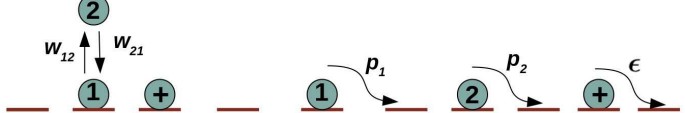

Figure 1: The figure illustrates all possible microscopic dynamical processes for the 2-TASEP-IAF with $\mu = 2$. The species 1 and species 2 particles can hop to right (if vacant) with rates $p_1$ and $p_2$ whereas the corresponding hopping rate for the impurity $(+)$ is $\epsilon$. The flip process between the species 1 and 2 can occur only in the presence of an impurity at the right neighbor. The corresponding flip rates are $w_{12}$ (for 1 transforming to 2) and $w_{21}$ (for 2 transforming to 1).

in Sec. 3. We discuss the behaviors of observables like species densities, drift current, flip current and spatial correlations from both analytical calculation and Monte Carlo simulations with variation of input parameters in Sec. 4. The $\mu$-TASEP-IAF is generalized to $\mu$-ASEP-IAF with bidirectional motions of the species in Sec. 5, where we show the corresponding matrix product states and discuss the negative differential mobility of particles. In Sec. 6, we summarize the results with future directions. We discuss the mapping between $\mu$-TASEP-IAF and multi lane TASEP in Appendix A. A variation of $\mu$-TASEP-IAF that comes up with better features in connection to traffic in multi-lane problems is discussed in Appendix B. The connections between $\mu$-TASEP-IAF and enzymatic chemical reactions are briefly presented in Appendix C. In Appendix D, we provide explicit solutions for the fugacity, in the grand canonical ensemble, for some special choices of the input parameters. The block-diagonal structure of the transition rate matrix dictating the transitions between configurations in the configuration space, is presented in Appendix E.

## 2    Model: $\mu$-TASEP-IAF

### 2.1    Microscopic dynamics

Let us consider a system of $\mu$ different species of particles and impurities on a one dimensional periodic lattice with $L$ sites $i = 1, 2, \ldots, L$. Each site can either be vacant or it can be occupied by only one particle of any of the species $I = 1, 2, \ldots, \mu$ or by an impurity. All the particles obey hardcore exclusion. The impurity and the vacancy are denoted by $+$ and 0, respectively. The system evolves according to the microscopic dynamics given below,

$$
\begin{aligned}
\text{drift (species)}: \quad & I0 \quad \xrightarrow{p_I} \quad 0I, \qquad I = 1, 2, ..., \mu, \\
\text{drift (impurity)}: \quad & +0, \quad \xrightarrow{\epsilon} \quad 0+ \\
\text{flip}: \quad & I+ \quad \underset{w_{KI}}{\overset{w_{IK}}{\rightleftharpoons}} \quad K+, \qquad I, K = 1, ..., \mu.
\end{aligned}
\tag{1}
$$

According to the dynamics of the 2-TASEP-IAF in Eq. (1), a particle of species $I$ can hop to its right nearest neighbor with rate $p_I$ if the target site is vacant. The impurity $(+)$ hopping rate is $\epsilon$. If the right neighbor of a particle of species $I$ is occupied by an impurity, then the species $I$ can transform to species $K$ with rate $w_{IK}$ and the reverse transformation from species $K$ to $I$ occurs with rate $w_{KI}$. Clearly, this flip dynamics is activated by the presence

of the impurities ($+$). The total number of impurities $N_+$ along with the total number of vacancies $N_0$ are conserved quantities, which can be readily seen from Eq. (1). The complete set of input parameters for the $\mu$-TASEP-IAF is $(p_I, \epsilon, w_{IK}, \rho_+, \rho_0)$, where $\rho_+ = N_+/L$ and $\rho_0 = N_0/L$ are the conserved densities for the impurities and the vacancies respectively. To illustrate the dynamics, we present a schematic figure of the allowed dynamical processes for the $\mu = 2$ case in Fig. 1.

From the microscopic dynamics in Eq. (1), it is clear that starting from a specific initial configuration, the different species and the impurities cannot overtake each other. Indeed the flip dynamics changes the number of accessible configurations by transforming one species to another, but does not allow the dynamics to be ergodic. To discuss the non-ergodicity with an example, let us consider an initial configuration (for 2-TASEP-IAF) of the form $\{\ldots 0 + 102011 + 2\ldots\}$. In the rest of this section, we will denote the particle under consideration by italics e.g. *1* for the chosen particle 1, *2* for the chosen particle 2 etc. If we consider the *1* in $\{\ldots 0 + 10201\mathit{1} + 2\ldots\}$, it can transform into *2* by the $+$ at its right neighbor, thereby changing the configuration to $\{\ldots 0 + 10201\mathit{2} + 2\ldots\}$. However, another *1* in the initial configuration $\{\ldots 0 + 1020\mathit{1}1 + 2\ldots\}$ can never transform to *2* at any stage of the evolution because it can never come in contact with any $+$ (due to the non-overtaking nature of the dynamics), so that the configuration $\{\ldots 0 + 1020\mathit{2}1 + 2\ldots\}$ is never accessible.

## 2.2 Steady state: matrix product ansatz

Any configuration of the $\mu$-TASEP-IAF can be represented by $\{s_i\} \equiv \{s_1, s_2, \ldots, s_L\}$, where $s_i$ denotes the occupation at site $i$. Clearly, $s_i$ can be one of the species $K = 1, 2, \ldots, \mu$ or it can be an impurity ($+$) or it can be a vacancy (0). We find that the steady state of the present model can be written in the following matrix product form

$$P(\{s_i\}) \propto \mathrm{Tr}\left[\prod_{i=1}^{L} X_i\right],$$

$$X_i = E\,\delta_{s_i,0} + A\,\delta_{s_i,+} + \sum_{K=1}^{\mu} D_K\,\delta_{s_i,K}. \tag{2}$$

In Eq. (2), any configuration $\{s_i\}$ is represented by a string of matrices $\{X_i\}$ where the matrices $D_K, A$ and $E$ corresponds to a particle of species $K$, impurity and vacancy respectively. The time evolution of any configuration of the $\mu$-TASEP-IAF is dictated by the Master equation

$$\frac{d}{dt}|P(t)\rangle = M|P(t)\rangle, \tag{3}$$

which in steady state becomes $M|P\rangle = 0$. Here $|P\rangle$ is a column vector containing all possible configurations and $M$ is the rate matrix made up of the transition rates between configurations. Since the dynamics in Eq. (1) is a two-site microscopic dynamics, the transition rate matrix, under the periodic boundary condition, can be expressed as

$$M = \sum_{i=1}^{L} \left(I \otimes \ldots I \otimes \mathcal{M}_{i,i+1} \otimes I \cdots \otimes I\right), \tag{4}$$

where $\mathcal{M}_{i,i+1}$ is a $(\mu+2)^2 \times (\mu+2)^2$ dimensional matrix and $I$ is $(\mu+2) \times (\mu+2)$ dimensional identity matrix placed at every site except the pair $(i, i+1)$. Then the steady state $M|P\rangle = 0$

of the $\mu$-TASEP-IAF can be achieved through the following two-site flux (probability current) cancellation condition

$$\mathcal{M}_{i,i+1}\mathbf{X}_i \otimes \mathbf{X}_{i+1} \;\;=\;\; \tilde{\mathbf{X}}_i \otimes \mathbf{X}_{i+1} - \mathbf{X}_i \otimes \tilde{\mathbf{X}}_{i+1}, \tag{5}$$

where

$$\mathbf{X} = (E, A, D_1, D_2 \ldots, D_\mu)^T, \tag{6}$$

and

$$\tilde{\mathbf{X}} = \left( \tilde{E}, \tilde{A}, \tilde{D}_1, \tilde{D}_2 \ldots, \tilde{D}_\mu \right)^T, \tag{7}$$

where $(.)^T$ denotes the transpose of the row vector $(.)$ and $\tilde{E}, \tilde{A}, \tilde{D}_K$ are auxiliary matrices that are introduced to satisfy the steady state equation and these have to be found out consistently along with the matrix representations for $E, D, D_K$ $(K = 1, 2, \ldots, \mu)$. We find that suitable choices for the auxiliary matrices for the $\mu$-TASEP-IAF are

$$\tilde{E} = 1, \;\; \tilde{A} = 0, \;\; \tilde{D}_K = 0 \quad K = 1, 2, \ldots, \mu. \tag{8}$$

Correspondingly, the matrices $E, A$ and $D_K$ have to obey the matrix algebra consisting of the equations given below

$$
\begin{aligned}
p_K D_K E &= D_K, & K &= 1, \ldots, \mu \\
\epsilon A E &= A, & & \\
\sum_{\substack{I=1 \\ I \neq K}}^{\mu} w_{IK} D_I A &= D_K A \sum_{\substack{I=1 \\ I \neq K}}^{\mu} w_{KI}, & K &= 1, \ldots, \mu.
\end{aligned}
\tag{9}
$$

The last relation in Eq. (9) is reminiscent of the Kirchhoff's current law for each species $K$, in the sense that the total flip current from all other species to species $K$ is equal to the total flip current from $K$ to all other species. Note that the matrix algebra in Eq. (9) allows scalar solutions when the hopping rates for every species and the impurity become equal i.e. $p_K = \epsilon$ for all $K$. Naturally for this special set of rates, since the matrices reduce to scalars, no spatial correlations exist between the constituents of the system. For any other choice of rates, we expect matrix solutions to the Eq. (9). Below we discuss the cases $\mu = 2$, $\mu = 3$ extensively with explicit matrix representations and then generalize them to get the matrix representations for general $\mu$.

$\underline{\mu = 2}$ : For the 2-TASEP-IAF $(K = 1, 2)$, the matrix algebra [Eq. (9)] simplifies to

$$
\begin{aligned}
p_1 D_1 E &= D_1, & p_2 D_2 E = D_2, \\
\epsilon A E &= A, & \\
w_{12} D_1 A &= w_{21} D_2 A.
\end{aligned}
\tag{10}
$$

Clearly, the matrix relation for the flip process becomes trivial for the two-species case implying the absence of net flip current between the two species. More precisely, the flip process satisfies detailed balance condition for $\mu = 2$. However, there are non-zero drift currents in

the system. We find the following $3 \times 3$ matrix representations that satisfy the matrix algebra in Eq. (10),

$$D_1 = w_{21} \begin{pmatrix} 1 & 1 & 0 \\ 0 & 0 & 0 \\ 0 & 0 & 0 \end{pmatrix}, \quad D_2 = w_{12} \begin{pmatrix} 1 & 0 & 1 \\ 0 & 0 & 0 \\ 0 & 0 & 0 \end{pmatrix},$$

$$E = \begin{pmatrix} \frac{1}{\epsilon} & 0 & 0 \\ \frac{1}{p_1} - \frac{1}{\epsilon} & \frac{1}{p_1} & 0 \\ \frac{1}{p_2} - \frac{1}{\epsilon} & 0 & \frac{1}{p_2} \end{pmatrix}, \quad A = \begin{pmatrix} 1 & 0 & 0 \\ 0 & 0 & 0 \\ 0 & 0 & 0 \end{pmatrix}. \tag{11}$$

The matrix representation of the impurity i.e. $A$, in the projector form, resembles that of the defect of second class particles in case of TASEP with first and second class particles [25, 71] except the fact that the matrices are infinite dimensional in Refs. [25, 71].

$\underline{\mu = 3 :}$ The matrix algebra in Eq. (9) for the 3-TASEP-IAF process reads as

$$\begin{aligned} p_1 D_1 E = D_1, \quad p_2 D_2 E &= D_2, \quad p_3 D_3 E = D_3, \\ \epsilon A E &= A, \\ w_{21} D_2 A + w_{31} D_3 A &= (w_{12} + w_{13}) D_1 A, \\ w_{12} D_1 A + w_{32} D_3 A &= (w_{21} + w_{23}) D_2 A, \\ w_{13} D_1 A + w_{23} D_2 A &= (w_{31} + w_{32}) D_3 A. \end{aligned} \tag{12}$$

In comparison to the last relation in Eq. (10), clearly the flip processes for the three-species case given by the last three relations in Eq. (12) do not require the detailed balance as a necessary condition. Rather, the general condition (without putting any constraint on the set of flip rates) that satisfies the flip processes in Eq. (12) is

$$w_{12} D_1 A - w_{21} D_2 A = w_{23} D_2 A - w_{32} D_3 A = w_{31} D_3 A - w_{13} D_1 A. \tag{13}$$

We obtain the following $4 \times 4$ representations of the matrices that satisfy Eq. (12) along with Eq. (13),

$$D_1 = d_1 \begin{pmatrix} 1 & 1 & 0 & 0 \\ 0 & 0 & 0 & 0 \\ 0 & 0 & 0 & 0 \\ 0 & 0 & 0 & 0 \end{pmatrix}, D_2 = d_2 \begin{pmatrix} 1 & 0 & 1 & 0 \\ 0 & 0 & 0 & 0 \\ 0 & 0 & 0 & 0 \\ 0 & 0 & 0 & 0 \end{pmatrix},$$

$$D_3 = d_3 \begin{pmatrix} 1 & 0 & 0 & 1 \\ 0 & 0 & 0 & 0 \\ 0 & 0 & 0 & 0 \\ 0 & 0 & 0 & 0 \end{pmatrix},$$

$$E = \begin{pmatrix} \frac{1}{\epsilon} & 0 & 0 & 0 \\ \frac{1}{p_1} - \frac{1}{\epsilon} & \frac{1}{p_1} & 0 & 0 \\ \frac{1}{p_2} - \frac{1}{\epsilon} & 0 & \frac{1}{p_2} & 0 \\ \frac{1}{p_3} - \frac{1}{\epsilon} & 0 & 0 & \frac{1}{p_3} \end{pmatrix}, A = \begin{pmatrix} 1 & 0 & 0 & 0 \\ 0 & 0 & 0 & 0 \\ 0 & 0 & 0 & 0 \\ 0 & 0 & 0 & 0 \end{pmatrix}, \tag{14}$$

where

$$d_1 = w_{21} w_{31} + w_{23} w_{31} + w_{32} w_{21},$$

$$
\begin{aligned}
d_2 &= w_{12}w_{32} + w_{13}w_{32} + w_{31}w_{12}, \\
d_3 &= w_{13}w_{23} + w_{12}w_{23} + w_{21}w_{13}.
\end{aligned}
\tag{15}
$$

We note that the condition Eq. (13), with the explicit matrix representations from Eq. (14), becomes

$$
w_{KI}D_K A - w_{IK}D_I A = \alpha A, \qquad \alpha = w_{12}w_{23}w_{31} - w_{21}w_{13}w_{32}.
\tag{16}
$$

The parameter $\alpha$ in Eq. (16) quantifies the deviations of the flip processes from the detailed balance condition between any pair of species. As we would see later, the net flip current between any two species is proportional to $\alpha$. In fact $\alpha = 0$, which puts some constraints on the flip-rates, correspond to a straightforward generalization of the two-species process to three-species process with similar flip process matrix relations $w_{KI}D_K A = w_{IK}D_I A$.

*general $\mu$ :* For the general case of $\mu$-TASEP-IAF ($K = 1, 2, \ldots, \mu$), the structures of the matrices $D_K, E, A$ would be similar to that of Eq. (11) and Eq. (14). More precisely, the matrices are $(\mu + 1) \times (\mu + 1)$ dimensional and the corresponding explicit representations of the matrices are given by

$$
\begin{aligned}
D_K &= d_K |1\rangle \left( \langle 1| + \langle K| \right), \quad K = 1, \ldots, \mu, \\
E &= \frac{1}{\epsilon} |1\rangle\langle 1| + \sum_{K=2}^{\mu+1} \frac{1}{p_{K-1}} |K\rangle\langle K| + \left( \frac{1}{p_{K-1}} - \frac{1}{\epsilon} \right) |K\rangle\langle 1|, \\
A &= |1\rangle\langle 1|.
\end{aligned}
\tag{17}
$$

In Eq. (17), the vector $\langle I| = (0, \ldots, 0, 1, 0, \ldots 0)$ where 1 is placed at the $I$-th element with all other elements being zero and $|I\rangle$ is the transpose of $\langle I|$. Notably, the values of the coefficients $d_K$ associated with matrices $D_K$ in Eq. (17), can be calculated by solving the set of $\mu$ homogeneous linear equations of the form

$$
d_K \sum_{I \neq K} w_{KI} + \sum_{I \neq K} w_{IK}d_I = 0, \quad K = 1, 2, \ldots, \mu.
\tag{18}
$$

The summations over the index $I$ in Eq. (18) generally includes all $I = 1, \ldots, \mu$ except $K$. This corresponds to the general dynamics in Eq. (1) where any two species can transform into one another in the presence of the impurity (at right neighbor). However, one might be interested in special cases where the flip processes are restricted between certain pairs of species only. For example, one particular situation can be where any species $K$ can only transform to species numbers $(K + 1)$ and $(K - 1)$. In that case, the parameter $\alpha$ [Eq. (16)] which dictates the flip-current between any two species, will be given by $\alpha = \left( \prod_{K=1}^{\mu} w_{KK+1} - \prod_{K=1}^{\mu} w_{K+1K} \right)$. Notably, this reduces to the general solution of $\mu = 3$ as given in Eq. (16). However, unlike the cases of $\mu = 2$ or $\mu = 3$ or the special instance of the multi-species case stated above which allow $\alpha$ to be independent of the species $I$, in general $\alpha$ would depend on the particulars of the pairs $(I, K)$ for $\mu > 3$. Mathematically, Eq. (16) would be generalized as follows

$$
w_{KI}D_K A - w_{IK}D_I A = \alpha_{KI} A, \quad I, K = 1, 2, 3, 4, \ldots, \mu,
\tag{19}
$$

where $\alpha_{KI}$ have different values for different pairs $(I, K)$. As an example, one can consider $\mu = 4$ with the allowed set of flip dynamics described in Fig. 2. In Fig. 2, among twelve total possible flip rates, only six are present. Specifically, in this example, the impurity cannot activate flips between species $(2, 3)$, implying $\alpha_{23} = 0$ whereas for other pairs $\alpha_{KI} \neq 0$.

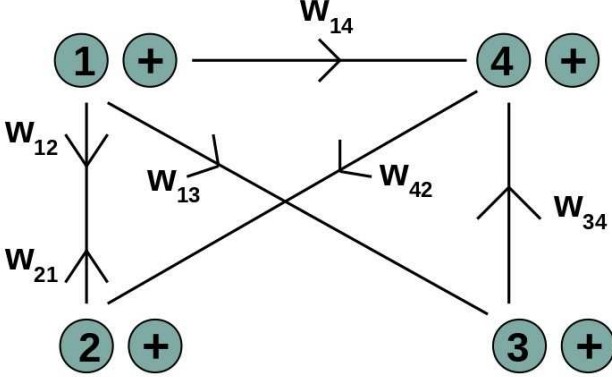

Figure 2: The figure illustrates an example of $\mu = 4$ case where certain flips are activated between pairs of species by the impurity, whereas some of the flips are absent e.g. $w_{24} = 0$. In this scenario, $\alpha_{KI}$ would be different for different pairs $(I, K)$.

## 3   Partition function for special initial configuration

The non-ergodic nature of the microscopic dynamics in Eq. (1) ensures that we cannot express the partition function of the $\mu$-TASEP-IAF in the usual form of $\text{Tr}[T^L]$, even under periodic boundary conditions. Here the "transfer matrix" $T$ refers to

$$T = z_0 E + z_+ A + \sum_K D_K, \tag{20}$$

with $z_0$ and $z_+$ being the fugacities corresponding to the vacancies and impurities respectively, in the grand canonical ensemble. This is because $T^L$ generates all configurations in the configuration space irrespective of the initial ordering of the species, but the dynamics in Eq. (1) allows only those configurations which preserve certain orderings from the initial configuration. To illustrate this with an example, let us consider a string of $(n + 1)$ number of species 1 particles followed by an impurity in the initial configuration i.e. $\{..11..11 + ..\} \equiv \{..1^{n+1} + ..\}$. At any step in the $\mu$-TASEP-IAF, the ordering of the first $n$ number of species 1 particles in this string cannot be broken, i.e. no species 2 particle or impurity can appear inside this string, except for the last 1 (which may transform to 2). On the other hand, $T^L$ generates configurations which do not preserve such orderings. Naturally, to calculate partition function for $\mu$-TASEP-IAF analytically, it becomes essential to choose suitable initial configurations for which we can correctly identify the accessible set of configurations in the steady state. In the rest of this section, we discuss such special initial configurations, corresponding steady states and partition functions.

### 3.1   $\mu = 2$

One special initial configuration $C(t = 0) \equiv C(0)$ (represented by matrices) for the 2-TASEP-IAF is

$$C(0) = \underbrace{D_1 A \ldots D_1 A} \; \underbrace{D_2 A \ldots D_2 A} \; \underbrace{D_2 \ldots D_2} \; \underbrace{D_1 \ldots D_1} \; \underbrace{E \ldots E}, \tag{21}$$

where $\underbrace{X...X}$ represents an uninterrupted sequence of the matrix $X$. We consider the densities of the uninterrupted sequences of $D_2$-s and $D_1$-s to be equal, which is $\bar{\rho} = \bar{N}/L$. Further, we have taken the two sequences of $D_1 A$ and $D_2 A$ to be of equal density $\rho_+$ so that the density of impurity ($A$) in each of these sequences is $\rho_+/2$. This ensures that the total density of impurities is $\rho_+$. The initial configuration in Eq. (21) satisfies the relation $\bar{\rho} = \frac{1}{2}(1 - \rho_0 - 2\rho_+)$, where $\rho_0$ and $\rho_+$ are densities of the vacancies and impurities respectively. In the steady state we have $\rho_0 + \rho_+ + \rho_1 + \rho_2 = 1$, with $\rho_1$ and $\rho_2$ being the average densities of species 1 and species 2 particles in the steady state. We emphasize that $\rho_0$ and $\rho_+$ are input parameters while $\rho_1$ and $\rho_2$ are derived quantities. Starting from Eq. (21), any accessible configuration $C_{ss}$ in the steady state is of the following generic form

$$C_{ss} = \prod_{k=1}^{N_+} (\tau D_1 + (1-\tau)D_2) E^{m_k} A E^{n_k} \prod_{i=1}^{\bar{N}} D_2 E^{r_i} \prod_{j=1}^{\bar{N}} D_1 E^{s_j}, \tag{22}$$

subjected to the constraint $\sum_{i=1}^{N_+}(m_i + n_i) + \sum_{j=1}^{\bar{N}}(r_j + s_j) = N_0$. The parameter $\tau$ in Eq. (22) can take value either 1 or 0. The partition function, obeying the above mentioned constraint, is given by

$$Q_{N_0, N_+} = \sum_{\{n_i\}} \sum_{\{m_i\}} \sum_{\{r_j\}} \sum_{\{s_j\}} \mathrm{Tr} \left[ \prod_{k=1}^{N_+}(D_1 + D_2) E^{m_k} A E^{n_k} \prod_{i=1}^{\bar{N}} D_2 E^{r_i} \prod_{j=1}^{\bar{N}} D_1 E^{s_j} \right]$$
$$\times \delta(\sum_{i=1}^{N_+}(n_i + m_i) + \sum_{j=1}^{\bar{N}}(r_j + s_j) - N_0). \tag{23}$$

It would be useful to get rid of the $\delta(.)$ constraint by associating a fugacity $z_0$ to the vacancy (represented by $E$) and considering the system in a grand canonical ensemble. The matrix strings $(D_1 + D_2) E^m A E^n$, $D_2 E^r$ and $D_1 E^s$ can be evaluated by incorporating the matrix algebra in Eq. (10) along with the explicit representations from Eq. (11). It should be mentioned that the projector form of $A$ [Eq. (11)] leads to factorization of the matrix strings, e.g.

$$\dots AD_1 E D_2 E A D_2 E E A \cdots = \dots |1\rangle \langle 1|D_1 E D_2 E|1\rangle \langle 1|D_2 E E|1\rangle \langle 1|\dots, \tag{24}$$

which helps significantly in carrying out the analytical calculations. Consequently, the partition function in the grand canonical ensemble under the periodic boundary condition, finally becomes

$$Q_{N_+}(z_0) = \left( \left[ \frac{w_{21}}{1 - \frac{z_0}{p_1}} + \frac{w_{12}}{1 - \frac{z_0}{p_2}} \right] \left( \frac{1}{1 - \frac{z_0}{\epsilon}} \right) \right)^{N_+} \left( \frac{w_{21} w_{12}}{(1 - \frac{z_0}{p_1})(1 - \frac{z_0}{p_2})} \right)^{\bar{N}}. \tag{25}$$

For the special initial configuration in Eq. (21), we have derived the partition function in Eq. (25). The fugacity $z_0$ can be obtained as a function of the vacancy density and other input parameters by inverting the density-fugacity relation

$$\rho_0 = \frac{z_0}{L} \frac{d}{dz_0} \ln(Q_{N_+}(z_0)). \tag{26}$$

In general, the solution of $z_0$ obtained from Eq. (26), using Mathematica, appears to be complicated and lengthy. However, in Appendix D, we would discuss two special cases (with

specific choices of the input parameter) that provide closed form solutions for the fugacity. The other conserved quantity, the impurity density is already fixed at $\rho_+ = N_+/L$. The expression Eq. (25) would be used for evaluating the observables of interest in the next Sec. 4.

### 3.2  $\mu = 3$

Similar to the initial configuration $C(t = 0)$ for $\mu = 2$, a suitable initial configuration for the three species case that enables us to perform analytical calculation of partition function and observables, is

$$C(0) \equiv \underbrace{D_1 A \ldots D_1 A}\ \underbrace{D_2 A \ldots D_2 A}\ \underbrace{D_3 A \ldots D_3 A}\underbrace{D_3 \ldots D_3}\ \underbrace{D_2 \ldots D_2}\ \underbrace{D_1 \ldots D_1}\ \underbrace{E \ldots E}. \quad (27)$$

The density of the uninterrupted sequence of each species $1, 2, 3$ in Eq. (27) is taken to be equal to $\bar{\rho} = \bar{N}/L$. Moreover, we have chosen the density of each of the sequences $D_1 A$, $D_2 A$ and $D_3 A$ to be $2\rho_+/3$ where the density of impurities in each of these sequences is $\rho_+/3$. Consequently, the choice of initial configuration in Eq. (27) ensures that the total density of impurities remain $\rho_+$. In the steady state, $\rho_0 + \rho_+ + \sum_{I=1}^{3} \rho_I = 1$, where $\rho_I$ is the average density of the non-conserved species $I$. Starting from Eq. (27), the form of any accessible configuration $C_{ss}$ in steady state would be

$$C_{ss} \equiv \prod_{k=1}^{N_+} \left(D_1 \delta_{\tau,1} + D_2 \delta_{\tau,2} + D_3 \delta_{\tau,3}\right) E^{m_k} A E^{n_k} \prod_{i=1}^{\bar{N}} D_3 E^{l_i} \prod_{i=1}^{\bar{N}} D_2 E^{r_i} \prod_{i=1}^{\bar{N}} D_1 E^{s_i}, \quad (28)$$

with the conservation of the total number of vacancies $N_0 = \sum_{k=1}^{N_+} (m_k + n_k) + \sum_{i=1}^{\bar{N}} (l_i + r_i + s_i)$, where $\delta_{\tau,K}$ is the Kronecker delta symbol with $K = 1, 2, 3$. As done in case of $\mu = 2$, here also we associate a fugacity $z_0$ with the vacancy. Using the matrix algebra from Eq. (12) alongside the matrix representations in Eqs.(14) and (15), the partition function in grand canonical ensemble becomes

$$Q_{N_+}(z_0) = \left(\left[\sum_{I=1}^{3} \frac{d_I}{1 - \frac{z_0}{p_I}}\right] \left(\frac{1}{1 - \frac{z_0}{\epsilon}}\right)\right)^{N_+} \left(\prod_{I=1}^{3} \frac{d_I}{1 - \frac{z_0}{p_I}}\right)^{\bar{N}}, \quad (29)$$

where the explicit expressions for $d_I$-s have been presented earlier in Eq. (15).

### 3.3  general $\mu$

For the $\mu$-TASEP-IAF ($K = 1, \ldots, \mu$), the generalization of the initial configurations in Eq. (21) ($\mu = 2$) and Eq. (27) ($\mu = 3$) would be

$$C(0) \equiv \left(\prod_{i=1}^{N_+/\mu} D_1 A \prod_{i=1}^{N_+/\mu} D_2 A \cdots \prod_{i=1}^{N_+/\mu} D_\mu A\right) \left(\prod_{i=1}^{\bar{N}} D_1 \prod_{i=1}^{\bar{N}} D_2 \cdots \prod_{i=1}^{\bar{N}} D_\mu\right) \prod_{i=1}^{N_0} E. \quad (30)$$

The above initial configuration is chosen in a way that the density of each sequence $D_I A$ $(I = 1, \ldots, \mu)$ is $2\rho_+/\mu$ in which the density of impurities is equal to $\rho_+/\mu$, so that the total impurity density adds up to $\rho_+$. We have $\rho_0 + \rho_+ + \sum_{I=1}^{\mu} \rho_I = 1$ in the steady state, with $\rho_I$ being the average density of the non-conserved species $I$. Proceeding in the same way as shown in cases of $\mu = 2$ and $\mu = 3$, we obtain the partition function for the general $\mu$-TASEP-IAF to be

$$Q_{N_+}(z_0) = \left( \left[ \sum_{I=1}^{\mu} \frac{d_I}{1 - \frac{z_0}{p_I}} \right] \left( \frac{1}{1 - \frac{z_0}{\epsilon}} \right) \right)^{N_+} \left( \prod_{I=1}^{\mu} \frac{d_I}{1 - \frac{z_0}{p_I}} \right)^{\bar{N}},$$

(31)

where $d_I$ is the solution of Eq. (18) and $z_0$ is the fugacity associated with the vacancy in the grand canonical ensemble.

In this section, we have derived the partition functions of the $\mu$-TASEP-IAF with $\mu = 2$, $\mu = 3$ and general $\mu$ in Eq. (25), Eq. (29) and Eq. (31), respectively, for specific initial configuration Eq. (21), Eq. (27) and Eq. (30), respectively, under periodic boundary conditions. These results would be useful to calculate the average values of observables in the next section for the same initial configurations discussed here.

## 4 Observables: comparisons of analytical results with Monte Carlo simulations

In this section, we analytically calculate the following observables in the steady state, (i) average density $\rho_I$ of the non-conserved species $I$, (ii) average drift currents $J_{I0}$ and $J_{+0}$, for species $I$ and impurities respectively, (iii) average flip current $J_{I \leftrightarrow K}$ between species pair $(I, K)$ and (iv) two-point correlations $C_{0I}$ between vacancies (0) and species $I$. Mostly we will restrict the calculations to the number of species $\mu = 2$, except using $\mu = 3$ for the case of average flip current (since there is no net flip current between pair of species for $\mu = 2$). In particular, starting from the special initial configuration Eq. (21) for $\mu = 2$ (or, Eq. (27) for $\mu = 3$), we will show agreements between the analytical calculations and the Monte Carlo simulation results.

### 4.1 Species densities

First we consider the average densities $(\rho_I)$ of the non-conserved species $I = 1, 2$. The formal expression for $\rho_I$ in the steady state under the periodic boundary condition, can be written as

$$\rho_1 = \frac{1}{2}(1 - \rho_0 - 2\rho_+) + \frac{\rho_+}{Q_{N_+}} \sum_{n_1=0}^{\infty} .. \sum_{n_{N_+}=0}^{\infty} \sum_{m_1=0}^{\infty} .. \sum_{m_{N_+}=0}^{\infty} \sum_{r_1=0}^{\infty} .. \sum_{r_{\bar{N}}=0}^{\infty} \sum_{s_1=0}^{\infty} .. \sum_{s_{\bar{N}}=0}^{\infty}$$

$$\text{Tr} \left[ D_1(z_0 E)^{m_1} A(z_0 E)^{n_1} \prod_{k=2}^{N_+} (D_1 + D_2)(z_0 E)^{m_k} A(z_0 E)^{n_k} \prod_{k=1}^{\bar{N}} D_2(z_0 E)^{r_k} \prod_{k=1}^{\bar{N}} D_1(z_0 E)^{s_k} \right].$$

(32)

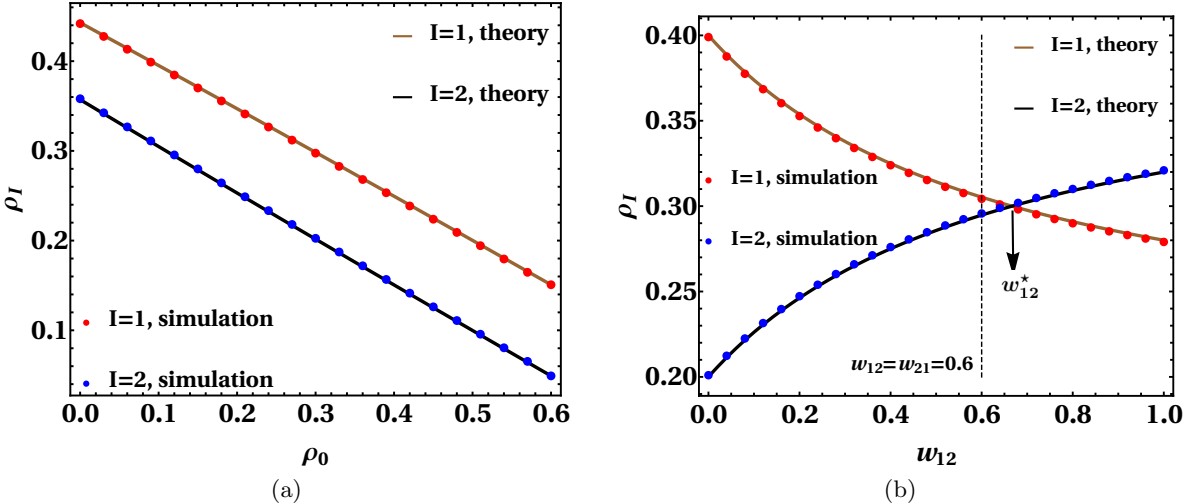

Figure 3: The figures (a) and (b) show the variations of the average species densities $\rho_I$ ($I = 1, 2$) against vacancy density $\rho_0$ and flip rate $w_{12}$ respectively in the steady state. The species densities decrease linearly with increasing $\rho_0$ in (a). In (b), with increasing $w_{12}$, $\rho_1$ and $\rho_2$ decrease and increase respectively, both in nonlinear manners. Notably, in the parameter range $w_{21} < w_{12} < w_{12}^\star$, we observe $\rho_2 < \rho_1$ in spite of the higher flip rate of transformation from species 1 to species 2. The common parameters for both figures (a) and (b) are $L = 10^3, p_1 = 0.3, p_2 = 1.0, \epsilon = 0.1, \rho_+ = 0.2$. For (a), $w_{12} = 0.4$ and $w_{21} = 1.0$. For (b), $\rho_0 = 0.2$ and $w_{21} = 0.6$. The ensemble average is done over $10^5$ samples.

To elaborate Eq. (32), the main point is to note the expression inside the trace (Tr[.]) that denotes configurations with at least one $D_1$. This can be understood more clearly by comparing it with the expression for any possible configuration in Eq. (22). From Eq. (22), to arrive at the matrix string inside the trace in Eq. (32), one has to put $\tau = 1$ for one $k$ value to ensure the presence of at least one species 1 particle ($D_1$) in the configuration. Since this $D_1$ could have been placed for any $k = 1, 2, \ldots, N_+$, we have a combinatorial pre-factor $\rho_+ = N_+/L$ in Eq. (32). The summations over all the variables $\{m, n, r, s\}$ from zero to infinity are performed as the system is considered in the grand canonical ensemble. The first factor $(1 - \rho_0 - 2\rho_+)/2$ in Eq. (32) is to take care of the $\bar{N}$ number of $D_1$-s present in the initial configuration Eq. (21) that cannot flip. Using the matrix algebra and matrix representations from Eqs. (10) and (11) respectively, we finally arrive at the following expressions for the average densities of the non-conserved species,

$$\rho_1 = \rho_+ \frac{w_{21}}{\left(1 - \frac{z_0}{p_1}\right)} \frac{1}{\left[\frac{w_{21}}{1 - \frac{z_0}{p_1}} + \frac{w_{12}}{1 - \frac{z_0}{p_2}}\right]} + \frac{1}{2}(1 - 2\rho_+ - \rho_0)$$

$$\rho_2 = \rho_+ \frac{w_{12}}{\left(1 - \frac{z_0}{p_2}\right)} \frac{1}{\left[\frac{w_{21}}{1 - \frac{z_0}{p_1}} + \frac{w_{12}}{1 - \frac{z_0}{p_2}}\right]} + \frac{1}{2}(1 - 2\rho_+ - \rho_0).$$

$$(33)$$

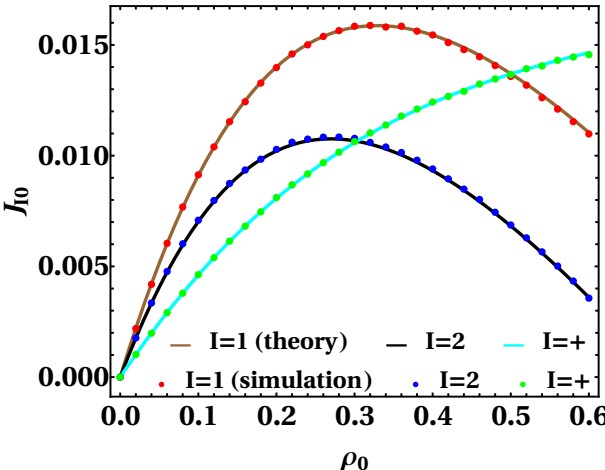

Figure 4: The figure illustrates the non-monotonic behaviors of the drift currents $J_{I0}$ ($I = 1, 2$) with increasing vacancy density $\rho_0$, unlike the drift current for the impurities $J_{+0}$ which increases monotonically with increasing $\rho_0$. The analytical results (solid lines) are in agreement with the Monte Carlo simulation results (points). The parameters used are $L = 10^3, p_1 = 0.3, p_2 = 1.0, \epsilon = 0.1, w_{12} = 0.4, w_{21} = 1.0, \rho_+ = 0.2$. The ensemble average is done over $10^5$ samples.

The fugacity $z_0$ is obtained in terms of the input parameters by solving the density-fugacity relation $\rho_0 = z_0 \frac{d}{dz_0} \ln(Q_{N_+}(z_0))$ using Mathematica. Replacing the solution of $z_0$ (which is too lengthy to provide here) in Eq. (33), we finally get the average densities as functions of input parameters $(p_{1,2}, w_{12,21}, \epsilon, \rho_0, \rho_+)$ only.

We compare the analytical results with those of Monte Carlo simulations, starting from the same initial configuration Eq. (21). In the simulation, we vary the vacancy density in the initial configuration by changing the lengths of the uninterrupted strings of $D_2$ and $D_1$ in Eq. (21) i.e. simply by tuning $\bar{\rho}$ which is related to vacancy density $\rho_0$ as $\rho_0 + 2\bar{\rho} = 1 - 2\rho_+$. In Figs. 3(a) and (b), we observe that the analytical and simulation results are in agreement with each other, where $\rho_I$ is plotted against $\rho_0$ and $w_{12}$ (flip rate of species 1 to species 2), respectively. The species densities decrease linearly with increasing $\rho_0$ [Fig. 3(a)] whereas they decrease in non-linear fashion with increasing $w_{12}$ [Fig. 3(b)]. Notably, in the absence of any drift, we would have $\rho_2 = \rho_1$ exactly at $w_{12}^\star = w_{21}$. However, due to the hopping process, this point shifts to

$$w_{12}^\star = w_{21} \frac{\left(1 - \frac{z_0}{p_2}\right)}{\left(1 - \frac{z_0}{p_1}\right)}. \tag{34}$$

Consequently in Fig. 3(b), for a particular set of chosen parameters, we observe that for $w_{21} < w_{12} < w_{12}^\star$, one still has $\rho_2 < \rho_1$. In other words, when $w_{12} \in (w_{21}, w_{12}^\star)$, although the species 1 particles more often transform to species 2 particles, still the average density of species 2 particles is less than that of species 1 particles. For any value of $\mu$, the general

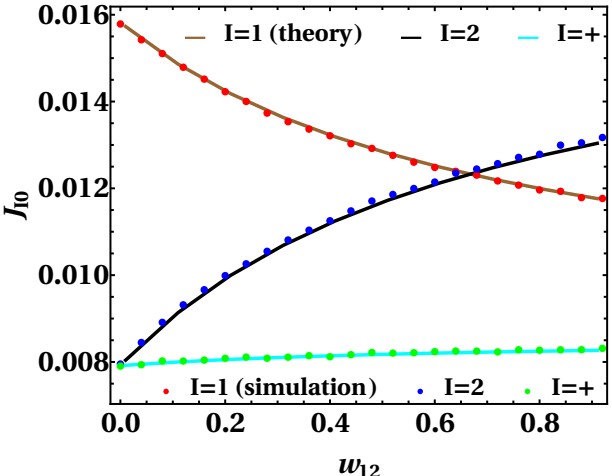

Figure 5: The figure shows comparison of drift currents $J_{I0}$ between theory (solid lines) and Monte Carlo simulations (points) by changing flip rate $w_{12}$. The nonlinear monotonic decrease and increase in $J_{10}$ and $J_{20}$ respectively are similar in nature to the effect of $w_{12}$ on the species densities (Fig. 3). The impurity drift current increases slowly with increasing $w_{12}$. The parameters used are $L = 10^3, p_1 = 0.3, p_2 = 1.0, \epsilon = 0.1, w_{21} = 0.6, \rho_+ = 0.2, \rho_0 = 0.2$. The ensemble average is done over $10^5$ samples.

expression for the average density $\rho_I$ for the non-conserved species $I$ ($I = 1, 2, \ldots, \mu$) is

$$\rho_I = \rho_+ \frac{d_I}{\left(1 - \frac{z_0}{p_I}\right)} \frac{1}{\sum_{K=1}^{\mu} \frac{d_K}{\left(1 - \frac{z_0}{p_K}\right)}} + \frac{1}{\mu}(1 - 2\rho_+ - \rho_0), \tag{35}$$

where $d_I$ is the solution of Eq. (18) (e.g. the solution for $\mu = 3$ is explicitly given in Eq. (15)).

### 4.2 Drift current

Next we consider the average drift currents $J_{I0}$ and $J_{+0}$ for the non-conserved species $I$ ($I = 1, 2$) and the impurity respectively. We focus on $I = 1$ to explain the procedure for calculating the current $J_{10}$, because the parallel procedure applies for any other species. The average drift current $J_{10}$ is equal to $p_1 \langle 10 \rangle$, where $\langle 10 \rangle$ is the ensemble average of the pair 10. In terms of matrices the expression $\langle 10 \rangle$ simply translates to $\langle D_1 E \rangle$. The current $J_{10}$ can be calculated in two parts,

$$J_{10} = p_1 \langle 10 \rangle = p_1 \langle D_1 E \rangle = J_{10}^{(1)} + J_{10}^{(2)} = p_1 \langle D_1 E \rangle^{(1)} + p_1 \langle D_1 E \rangle^{(2)}, \tag{36}$$

where $J_{10}^{(1)} = p_1 \langle D_1 E \rangle^{(1)}$ is the contribution from the drift of species 1 particles that can flip and $J_{10}^{(2)} = p_1 \langle D_1 E \rangle^{(2)}$ is the corresponding contribution from species 1 particles that cannot flip (as they cannot have any impurity as right neighbor) according to the initial configuration in Eq. (21). Correspondingly, the term $D_1 E$ in the averages $\langle D_1 E \rangle^{(1)}$ and $\langle D_1 E \rangle^{(2)}$ would come from the product sequences $(\tau D_1 + (1 - \tau)D_2)E^m A$ and $D_1 E^s$ respectively Eq. (22).

The expression for $J_{10}^{(1)}$ is given by

$$J_{10}^{(1)} = p_1 \langle D_1 E \rangle^{(1)} = \frac{\rho_+}{Q_{N_+}} \sum_{n_1=0}^{\infty} .. \sum_{n_{N_+}=0}^{\infty} \sum_{m_1=1}^{\infty} .. \sum_{m_{N_+}=0}^{\infty} \sum_{r_1=0}^{\infty} .. \sum_{r_{\bar{N}}=0}^{\infty} .. \sum_{s_{\bar{N}}=0}^{\infty}$$

$$\mathrm{Tr}\left[ z_0 p_1 D_1 E(z_0 E)^{m_1-1} A(z_0 E)^{n_1} \prod_{k=2}^{N_+} (D_1 + D_2)(z_0 E)^{m_k} A(z_0 E)^{n_k} \prod_{k=1}^{\bar{N}} D_2(z_0 E)^{r_k} \prod_{k=1}^{\bar{N}} D_1(z_0 E)^{s_k} \right].$$

$$(37)$$

The construction of Eq. (37) follows similar arguments as of Eq. (32), except now we have to place $D_1 E$ instead of $D_1$. This also reflects in the summations, note that the lower limit of the index $m_1$ has been changed to 1 instead of 0 to ensure the presence of one $D_1 E$. Similarly, the formal expression for $J_{10}^{(2)}$ is

$$J_{10}^{(2)} = p_1 \langle D_1 E \rangle^{(2)} = \frac{1}{Q_{N_+}} \sum_{n_1=0}^{\infty} .. \sum_{r_{\bar{N}}=0}^{\infty} \sum_{s_1=1}^{\infty} .. \sum_{s_{\bar{N}}=0}^{\infty}$$

$$\mathrm{Tr}\left[ \prod_{k=1}^{N_+} (D_1 + D_2)(z_0 E)^{m_k} A \prod_{k=1}^{\bar{N}} D_2(z_0 E)^{r_k} z_0 p_1 D_1 E(z_0 E)^{s_1-1} \prod_{k=2}^{\bar{N}} D_1(z_0 E)^{s_k} \right]. \quad (38)$$

Obviously in Eq. (38), the lower limit of the index $s_1$ is shifted to 1 from 0. Using the matrix algebra and matrix representations from Eqs. (10) and (11), we get from Eqs. (37) and (38):

$$J_{10}^{(1)} = z_0 \rho_+ \frac{w_{21}}{\left(1 - \frac{z_0}{p_1}\right)} \frac{1}{\left[\frac{w_{21}}{1 - \frac{z_0}{p_1}} + \frac{w_{12}}{1 - \frac{z_0}{p_2}}\right]}, \qquad J_{10}^{(2)} = \frac{z_0}{2}(1 - 2\rho_+ - \rho_0). \qquad (39)$$

Substituting Eq. (39) into Eq. (36), we obtain $J_{10}$. Following the same procedures, one can calculate $J_{20}$ and $J_{+0}$. We finally arrive at the analytical expressions for the drift currents under the periodic boundary condition, given below:

$$
\begin{aligned}
J_{10} &= z_0 \rho_+ \frac{w_{21}}{\left(1 - \frac{z_0}{p_1}\right)} \frac{1}{\left[\frac{w_{21}}{1 - \frac{z_0}{p_1}} + \frac{w_{12}}{1 - \frac{z_0}{p_2}}\right]} + \frac{z_0}{2}(1 - 2\rho_+ - \rho_0), \\
J_{20} &= z_0 \rho_+ \frac{w_{12}}{\left(1 - \frac{z_0}{p_2}\right)} \frac{1}{\left[\frac{w_{21}}{1 - \frac{z_0}{p_1}} + \frac{w_{12}}{1 - \frac{z_0}{p_2}}\right]} + \frac{z_0}{2}(1 - 2\rho_+ - \rho_0), \\
J_{+0} &= \rho_+ z_0.
\end{aligned}
\qquad (40)
$$

In Figs. 4 and 5 we present the variation of the drift currents as functions of vacancy density $\rho_0$ and flip rate $w_{12}$, respectively. For both cases, the analytical results match with the Monte Carlo simulation results. The drift currents for the species 1 and 2 exhibit non-monotonic behaviors with increasing vacancy density whereas the impurity drift current increases monotonically (Fig. 4). At lower vacancy densities, as we increase $\rho_0$, the chances for hopping increase, thereby increasing the drift current in Fig. 4. However, after a particular value of $\rho_0$, if we increase it further, the densities of the non-conserved species fall considerably so that drift current ultimately decreases, although there are many vacancies in the system. Since the impurities do not flip, the density of impurities is fixed and the corresponding impurity

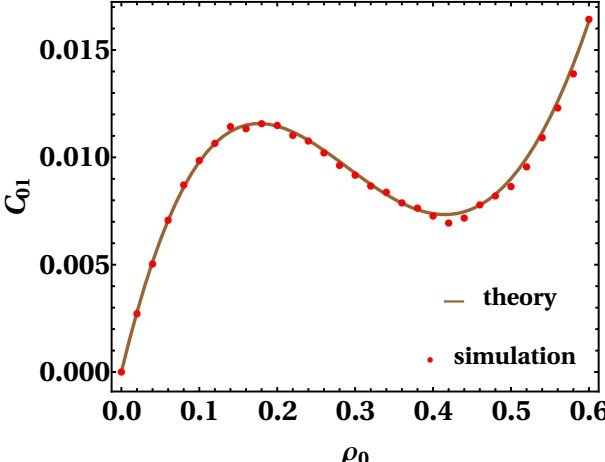

Figure 6: The figure illustrates the variation of two-point correlation $C_{01}$ between the vacancy and species 1 by tuning the vacancy density $\rho_0$. We observe that this correlation initially starts increasing with increasing $\rho_0$, reaches to a local maximum, then decreases. However, as $\rho_0$ is increased further, $C_{01}$ reaches to a local minimum and then starts increasing again. The parameters used are $L = 10^3, p_1 = 0.3, p_2 = 1.0, \epsilon = 0.1, w_{12} = 0.8, w_{21} = 1.0, \rho_+ = 0.2$. The ensemble average is done over $10^5$ samples.

drift current can only increase with increasing $\rho_0$ (Fig. 4). The maximum of the drift current for different species generally occur at distinct values of $\rho_0$. With variation of the flip rate $w_{12}$ (Fig. 5), the drift currents of the non-conserved species show similar non-linear behaviors like their corresponding densities [Fig. 3(b)]. Notably, although the flip dynamics does not affect the drift of the impurities explicitly, still we observe that the impurity drift current increases with increasing $w_{12}$, albeit weakly.

For any positive integer value of $\mu$, the general expression for the drift current of any non-conserved species $I$ $(I = 1, \ldots, \mu)$ is obtained to be

$$J_{I0} = z_0 \rho_+ \frac{d_I}{\left(1 - \frac{z_0}{p_I}\right)} \frac{1}{\sum\limits_{K=1}^{\mu} \frac{d_K}{\left(1 - \frac{z_0}{p_K}\right)}} + \frac{z_0}{\mu}(1 - 2\rho_+ - \rho_0), \tag{41}$$

where the $d_I$ is the solution of Eq. (18).

## 4.3 Two-point correlations

Besides the currents, we would like to calculate some other two point functions which have interesting features. We have basically calculated the two-point functions $\langle 10 \rangle$ and $\langle 20 \rangle$ in the process of determining the drift currents $J_{10}$ and $J_{20}$ respectively. Now we focus on the nearest neighbors two-point correlations involving $\langle 01 \rangle$ and $\langle 02 \rangle$. We find the exact expressions for the corresponding two-point correlations under the periodic boundary condition as

$$C_{01} = \langle 0\,1 \rangle - \langle 0 \rangle \langle 1 \rangle = \left(\frac{z_0}{\epsilon} - \rho_0\right) \frac{w_{21}}{\left(1 - \frac{z_0}{p_1}\right)} \frac{\rho_+}{\left[\frac{w_{21}}{1 - \frac{z_0}{p_1}} + \frac{w_{12}}{1 - \frac{z_0}{p_2}}\right]}$$

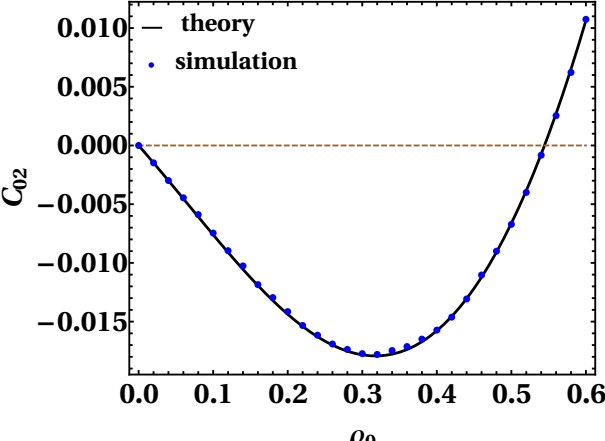

Figure 7: The figure shows a crossover from negative correlation to positive correlation for $C_{02}$ (between vacancy and species 2) with increasing vacancy density $\rho_0$. There is some special intermediate density for which $C_{02}$ becomes zero. The parameters used are $L = 10^3, p_1 = 0.3, p_2 = 1.0, \epsilon = 0.1, w_{12} = 0.8, w_{21} = 1.0, \rho_+ = 0.2$. The ensemble average is done over $10^5$ samples.

$$
\begin{aligned}
C_{02} = \langle 0\,2 \rangle - \langle 0 \rangle \langle 2 \rangle \;\; = \;\; & + \left( \frac{z_0}{2p_1} - \frac{\rho_0}{2} \right) (1 - 2\rho_+ - \rho_0), \\
& \left( \frac{z_0}{\epsilon} - \rho_0 \right) \frac{w_{12}}{\left(1 - \frac{z_0}{p_2}\right)} \frac{\rho_+}{\left[ \frac{w_{21}}{1 - \frac{z_0}{p_1}} + \frac{w_{12}}{1 - \frac{z_0}{p_2}} \right]} \\
& + \left( \frac{z_0}{2p_2} - \frac{\rho_0}{2} \right) (1 - 2\rho_+ - \rho_0).
\end{aligned}
\tag{42}
$$

The correlation $C_{01}$ is plotted against the vacancy density $\rho_0$ in Fig. 6. As $\rho_0$ is increased starting from zero, the correlation also increases and reaches a local maximum, followed by a decrease and reaching a local minimum. After this point, if the vacancy density is increased further, $C_{01}$ increases again. So, instead of a single maximum or single minimum, the correlation $C_{01}$ interestingly exhibits both local maximum and local minimum with the variation of $\rho_0$. In other words, $C_{01}$ increases with increasing vacancy density for both sufficiently high and sufficiently low values of $\rho_0$, with intermediate non-monotonic character. In Fig. 7, it is interesting to see that the non-monotonic behavior of $C_{02}$ is such that it goes from negative correlation values to positive correlation values. Naturally, there exists some intermediate value of $\rho_0$ for which the special arrangements of the accessible steady state configurations makes the average correlation $C_{02}$ to be zero.

## 4.4 Flip current

All the observables we have discussed up to now (average species densities, drift currents, correlations), correspond to 2-TASEP-IAF. However, the net flip current is zero for $\mu = 2$. Therefore, in order to have a net non-zero flip current between pairs of species, here we consider the case $\mu = 3$ ($I = 1, 2, 3$). We denote the net flip current between species $I$ and $K$ as $J_{I \leftrightarrow K}$. For $\mu = 3$, we find that the net flip current between any two species $[(1, 2), (2, 3), (3, 1)]$ are equal to each other (i.e. independent of the indices $I$ and $K$) and its exact form is

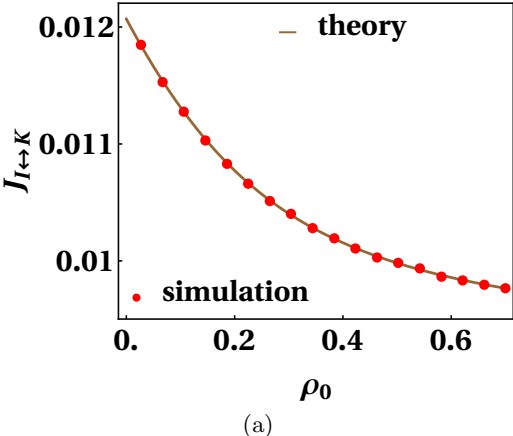
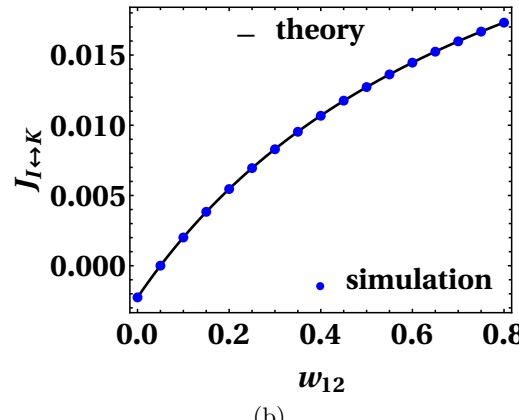

(a) (b)

Figure 8: The figures exhibit the variation of the flip current $J_{I\leftrightarrow K}$ with vacancy density $\rho_0$ and flip rate $w_{12}$ in (a) and (b), respectively. The analytical and Monte Carlo simulation results show good agreement. The flip current monotonically decreases with increasing $\rho_0$ whereas it increases monotonically with increasing $w_{12}$. The common parameters for both figures (a) and (b) are $L = 10^3, p_1 = 0.3, p_2 = 1.0, p_3 = 1.0, \epsilon = 0.1, w_{21} = 0.5, w_{23} = 0.5, w_{32} = 0.2, w_{31} = 0.8, w_{13} = 0.2, \rho_+ = 0.15$. For (a), $w_{12} = 0.4$ and for (b), $\rho_0 = 0.22$. The ensemble average is done over $10^5$ samples.

given by

$$J_{I\leftrightarrow K} = w_{IK}\langle I+\rangle - w_{KI}\langle K+\rangle = \rho_+ \frac{(w_{12}w_{23}w_{31} - w_{21}w_{13}w_{32})}{\left[\frac{d_1}{1-\frac{z_0}{p_1}} + \frac{d_2}{1-\frac{z_0}{p_2}} + \frac{d_3}{1-\frac{z_0}{p_3}}\right]}. \tag{43}$$

The initial configuration that we have used to arrive at Eq. (44) is the one in Eq. (27). In Eq. (44), we have calculated the current in the cyclic ordering i.e. $(I = 1, K = 2)$, $(I = 2, K = 3)$, $(I = 3, K = 1)$. We have presented the behavior of the flip current as functions of the vacancy density $\rho_0$ and flip rate $w_{12}$ in Figs. 8(a) and 8(b) respectively. We observe that the analytical calculation are in agreement with the Monte Carlo simulation results. In Fig. 8(a), the flip-current decreases monotonically in a nonlinear manner with increasing vacancy density. The reason behind this, is the decrease in species densities with increasing $\rho_0$ [Fig. 3(a)]. On the other hand, flip current between any pair of species increases monotonically with increasing flip rate $w_{12}$ (Fig. 8(b)).

The generalization of the formula Eq. (44) for any $\mu$, under the periodic boundary condition, is obtained as

$$J_{I\leftrightarrow K} = \rho_+ \frac{(d_I w_{IK} - d_K w_{KI})}{\sum\limits_{K=1}^{\mu} \frac{d_K}{\left(1-\frac{z_0}{p_K}\right)}}, \tag{44}$$

where $d_I, d_K$ are the solutions of Eq. (18). We should mention that, for $\mu > 3$, the flip currents between different pairs of species would be in general distinct from one another.

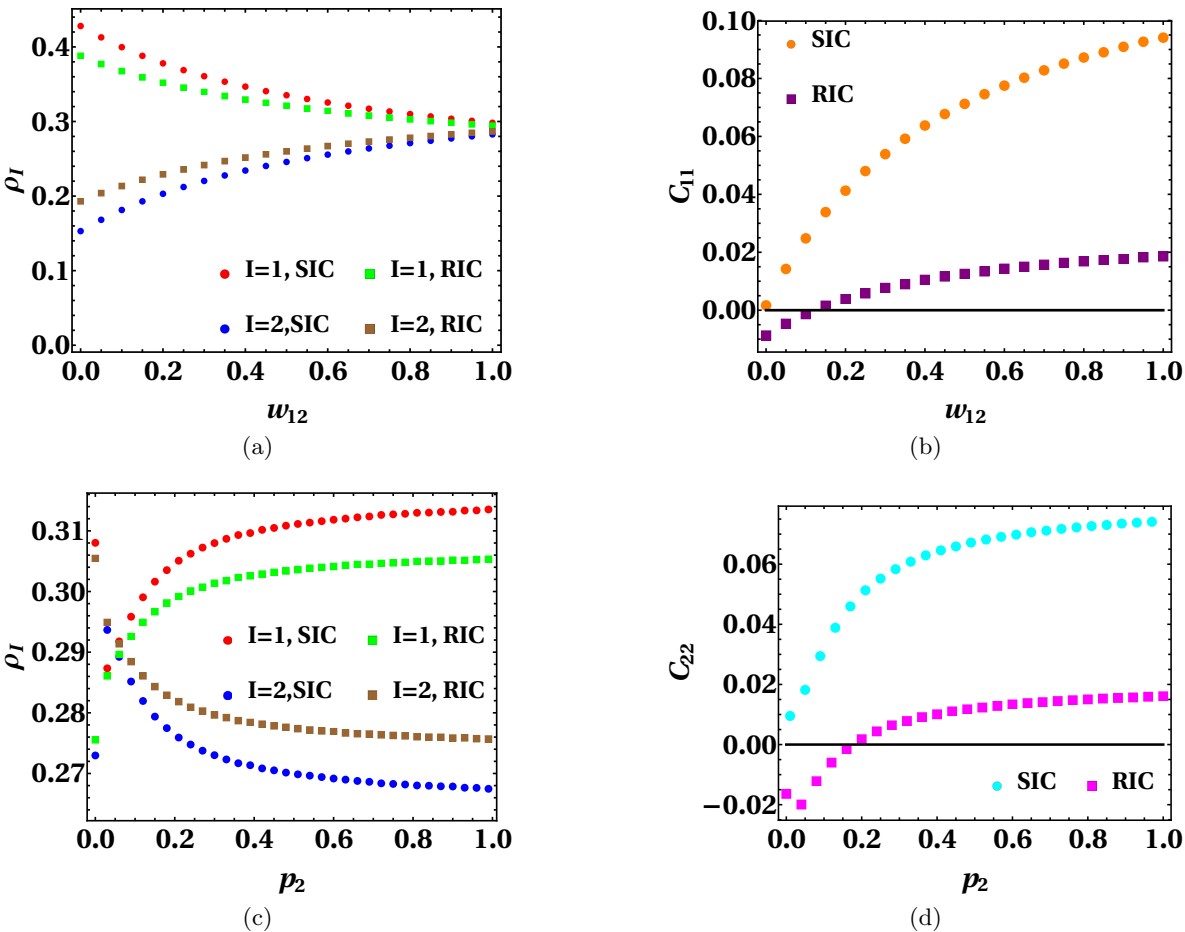

Figure 9: The set of figures exhibits the dependence of steady state values of observables on the choice of initial configuration. We observe clear deviations between SIC (denoted by circles) and RIC (denoted by rectangles) in (a)-(d) and the amount of deviation in each figure changes with the variation of input parameter. The common set of parameters used for (a)-(d) are $L = 10^3, p_1 = 0.3, \epsilon = 0.1, \rho_+ = 0.216, \rho_0 = 0.203, \rho_1(0) = 0.321, \rho_2(0) = 0.260$. The other parameters for (a), (b) are $p_2 = 1.0$, $w_{21} = 0.6$ and for (c), (d) are $w_{12} = 0.5, w_{21} = 0.4$. The ensemble averages are done over $10^5$ samples.

## 4.5 Non-ergodicity: dependence on initial configuration

Here we establish the non-ergodicity of the $\mu$-TASEP-IAF by showing explicitly the dependence of the average steady state values of observables on the choice of the initial configuration. For simplicity, we restrict ourselves to the case of $\mu = 2$. We choose two different initial configurations as follows. (i) The initial configuration given in Eq. (21) for which we know exactly which constituent (any species or impurity or vacancy) is placed at a given lattice site. Since the initial arrangement is specified completely, we call this configuration *specified initial configuration* (SIC). (ii) A random initial configuration where the constituent at each site is selected randomly such that the densities of impurities and vacancies, and the initial densities of species $I$ ($I = 1, 2$), are exactly the same as that of the SIC described in (i). Since the initial arrangement for this configuration is randomly carried out, we call it *random initial configuration* (RIC). To clarify, both SIC and RIC are characterized by the same set of rates $(p_1, p_2, \epsilon, w_{12}, w_{21})$ and the densities $(\rho_+, \rho_0, \rho_1(0), \rho_2(0))$, where $\rho_1(0), \rho_2(0)$ represent the initial ($t = 0$) densities of the non-conserved species 1 and 2 respectively. Although the analysis of the steady state for the SIC can be performed exactly as discussed already, the same could not be done for the RIC. Therefore, in this section we use Monte Carlo simulations to compare the steady state observable values for SIC and RIC.

We compare the steady state observable values for SIC and RIC with the same set of input parameters $(p_1, p_2, \epsilon, w_{12}, w_{21}, \rho_+, \rho_0)$ and same initial densities of the species $(\rho_1(0), \rho_2(0))$ in Fig. 9. We denote the data points for SIC and RIC with different symbols, circles and rectangles respectively, in Fig. 9. The variations of the non-conserved species densities $\rho_I$ are presented in Figs. 9(a) and 9(c) as functions of flip rate $w_{12}$ and hop rate $p_2$, respectively. Both figures exhibit clear quantitative differences between the density values for SIC and RIC. We observe that the deviations between SIC and RIC decreases (increases) with increasing $w_{12}$ ($p_2$). The initial configuration dependence of the steady state values of nearest neighbor two-point correlations are shown in Figs. 9(b) and (d). In Fig. 9(b), we observe that the correlation between species 1 particles $C_{11} = \langle 11 \rangle - \rho_1^2$ has distinct numerical values for SIC and RIC when the parameter $w_{12}$ is tuned. Interestingly, the correlation corresponding to RIC changes from negative to positive whereas the same for SIC remains positive with increasing $w_{12}$. This implies the existence of some intermediate $w_{12}$ which corresponds to uncorrelated species 1 particles for RIC, whereas they are correlated for the SIC. Similar kind of interesting behavior is observed for the correlation between species 2 particles $C_{22} = \langle 22 \rangle - \rho_2^2$ when plotted against $p_2$ in Fig. 9(d). Thus we have illustrated the dependence of steady state values of species densities and correlations on the choice of the initial configuration. The same can also be investigated in other two point and higher point functions.

We end this section with a general comment regarding the non-ergodicity in the present model. If we consider a sequence of the form $\{+s_i s_{i+1} \ldots s_n s_{n+1}+\}$ in an initial configuration, where $s_j = 0, 1, \ldots \mu$ but $s_j \neq +$ for $i \leqslant j \leqslant (n+1)$, then the ordering of different species $1, \ldots, \mu$ (not vacancies) for $i \leqslant j \leqslant n$ remains intact for the allowed subspace of configurations in the steady state. Naturally, number of such orderings increase with system size. Recent study [72] in context of classical reversible cellular automaton shows the number of local conservation laws increase exponentially with system size, leading to block diagonal form of the propagator with exponential scaling of the number of blocks with system size. In fact there are quantum systems like certain Lindbladian for quantum ASEP [73], dipole-conserving Hamiltonian [74] etc. for which the space of operators or states fragment into invariant subspaces whose number again scale exponentially with system size. It would be

interesting to investigate in future how does the number of conserved orderings scale with system size in our non-ergodic model, the detailed block diagonal structure of the transition rate matrix [$M$ in Eq. (3)] and the role of corresponding underlying symmetries. An explicit illustration of the block-diagonal structure of the rate matrix in the $\mu$-ASEP-IAF, for small system sizes, is presented in Appendix E.

# 5    Partially asymmetric generalization: $\mu$-ASEP-IAF

In this section, we consider the $\mu$-ASEP-IAF under periodic boundary conditions, a generalization of the $\mu$-TASEP-IAF in Eq. (1), by including partially asymmetric motions of different species of particles. A particle of species $I$ can hop towards right with rate $p_I$ and it can hop towards left with rate $q_I$ ($I = 1, \ldots, \mu$), if the target site is empty. Notably, the impurities are not allowed to hop to left. This naturally adds another way to distinguish the conserved impurities from all non-conserved species. The microscopic dynamics is given by,

$$
\begin{aligned}
\text{drift (species)}: \quad & I0 \quad \overset{p_I}{\underset{q_I}{\rightleftharpoons}} \quad 0I \quad I = 1, 2, ..., \mu \\
\text{drift (impurity)}: \quad & +0 \quad \overset{\epsilon}{\longrightarrow} \quad 0+ \\
\text{flip}: \quad & I+ \quad \overset{w_{IK}}{\underset{w_{KI}}{\rightleftharpoons}} \quad K+ \ \ I, K = 1, ..., \mu.
\end{aligned} \tag{45}
$$

The $\mu$-ASEP-IAF remains non-ergodic in nature. Since we could obtain the steady state of the $\mu$-TASEP-IAF using matrix product ansatz [Eq. (2)], we assume the same can be done for the partially asymmetric motion also.

## 5.1    Matrix algebra, auxiliaries and matrix representations

The matrix algebra for the dynamics in Eq. (45) under the periodic boundary condition, is

$$
\begin{aligned}
p_K D_K E - q_K E D_K &= D_K, & K = 1, \ldots, \mu \\
\epsilon A E &= A, & \\
\sum_{\substack{I=1 \\ I \neq K}}^{\mu} w_{IK} D_I A &= D_K A \sum_{\substack{I=1 \\ I \neq K}}^{\mu} w_{KI}, & K = 1, \ldots, \mu.
\end{aligned} \tag{46}
$$

In comparison to the matrix algebra [Eq. (9)] for the $\mu$-TASEP-IAF, the only changes occurring in Eq. (46) correspond to the drifts of the non-conserving species. At this point, we should mention that the matrix equation $p_K D_K E - q_K E D_K = D_K$ has been studied in Ref. [55], in context of a conserved disordered ASEP model. Due to the presence of the impurities and the flip processes activated by them, the matrix algebra for $\mu$-ASEP-IAF in Eq. (46), can be considered as a generalization of the matrix algebra in Ref. [55]. To arrive at the matrix algebra in Eq. (46) from the dynamics (45), ansatz (2) and flux cancellation condition (5), the choice of the auxiliary matrices are the same as the totally asymmetric case, i.e.

$$
\tilde{E} = 1, \ \tilde{A} = 0, \ \tilde{D}_K = 0 \quad K = 1, 2, \ldots, \mu. \tag{47}
$$

However, unlike the totally asymmetric case, we find the matrix representations for the $\mu$-ASEP-IAF to be infinite dimensional. Notably, this does not necessarily eliminate the possibility of getting alternate finite dimensional representations of the matrices. Below we present

the matrix representations for $\mu = 3$ case explicitly (as we will stick to $\mu = 3$ for the discussion of observable in this section) and mention the changes required to construct the matrices for any $\mu > 0$.

$\underline{\mu = 3}$ : A possible set of representations of the matrices for the 3-ASEP-IAF ($I = 1, 2, 3$) is the following

$$
E = \begin{pmatrix}
0 & 0 & 0 & 0 & . & . \\
1 & 0 & 0 & 0 & . & . \\
0 & 1 & 0 & 0 & . & . \\
0 & 0 & 1 & 0 & . & . \\
0 & 0 & 0 & 1 & & \\
. & . & & & & . \\
. & . & & & . &
\end{pmatrix}, \quad
A = \begin{pmatrix}
1 & \frac{1}{\epsilon} & \frac{1}{\epsilon^2} & \frac{1}{\epsilon^3} & . & . \\
0 & 0 & 0 & 0 & . & . \\
0 & 0 & 0 & 0 & . & . \\
. & . & . & . & . & . \\
. & . & . & . & . & .
\end{pmatrix}
$$

$$
D_I = \begin{pmatrix}
d_I^{1,1} & d_I^{1,2} & d_I^{1,3} & d_I^{1,4} & . & . \\
0 & d_I^{2,2} & d_I^{2,3} & d_I^{2,4} & . & . \\
0 & 0 & d_I^{3,3} & d_I^{3,4} & . & . \\
0 & 0 & 0 & d_I^{4,4} & . & . \\
. & . & & & . & \\
. & . & & & & .
\end{pmatrix}, \quad I = 1, 2, 3
$$

$$
d_I^{m,m+r} = \frac{(m)_r}{r! \, p_I^r} \left( \frac{q_I}{p_I} \right)^{m-1} d_I^{1,1}, \qquad \forall r \geqslant 0
$$

$$
d_1^{1,1} = w_{21} w_{31} + w_{23} w_{31} + w_{32} w_{21},
$$
$$
d_2^{1,1} = w_{12} w_{32} + w_{13} w_{32} + w_{31} w_{12},
$$
$$
d_3^{1,1} = w_{13} w_{23} + w_{12} w_{23} + w_{21} w_{13}. \tag{48}
$$

In the absence of the impurities ($A$) and the flip processes, the term $d_\mu^{1,1}$ for every $\mu$ becomes unity, and corresponding the matrix representations for $D_I$ and $E$ in Eq. (48) (and their generalizations for general $\mu$) are the same as that of the conserved disordered ASEP [27, 55]. In Eq. (48), we observe that the matrices ($D_I$) corresponding to the species ($I$) are upper triangular. The subscript $I$ in matrix element $d_I^{i,j}$ denotes the species $I$ whereas the superscript $(i, j)$ refers to the $i$-th row and $j$-th column of the matrix. The notation $(m)_r$ used in the expression of $d_I^{m,m+r}$ corresponds to the Pochhammer symbol for rising factorials, $(m)_r = m(m+1)(m+2)\ldots(m+r-1)$ with $(m)_0 = 1$. The matrix $E$ representing vacancy is a lower shift matrix and the matrix $A$ representing impurity has non-zero terms in a single row only. For the simpler case $\mu = 2$ the only changes in comparison to Eq. (48) will be in the values of $d_I^{1,1}$ ($I = 1, 2$), which would be simply $d_1^{1,1} = w_{21}$ and $d_2^{1,1} = w_{12}$. In fact, the matrix representations [Eq. (48)] for the 3-ASEP-IAF can be generalized for any $\mu > 0$ in a straightforward manner. The representations will remain the same, only the values of $d_I^{1,1}$ ($I = 1, 2, \ldots, \mu$) would change where $d_I^{1,1}$ is the solution of Eq. (18).

## 5.2 Partition function for special initial configuration

In the totally asymmetric case, for general $\mu$, we have considered the specific initial configuration Eq. (30) which leads us to acquire analytical expressions for observables of interest. We choose a particular case of Eq. (30), namely the $\bar{\rho} = 0$ case, as the special initial configuration for analytical calculation in $\mu$-ASEP-IAF. More precisely, the choice of our special

initial configuration for $\mu$-ASEP-IAF is,

$$C(0) \equiv \prod_{i=1}^{N_+/\mu} D_1 A \prod_{i=1}^{N_+/\mu} D_2 A \cdots \prod_{i=1}^{N_+/\mu} D_\mu A \prod_{i=1}^{N_0} E. \tag{49}$$

The initial configuration Eq. (49) is chosen in a way that fixes the total impurity density to $\rho_+$. In comparison to Eq. (30), the initial configuration in Eq. (49) is simpler and does not contain consecutive $D_I$-s. We would see shortly that this specific choice is sufficient to show negative differential mobility in $\mu$-ASEP-IAF. We find the partition function in the steady state under the periodic boundary condition corresponding to the initial configuration Eq. (49), to be

$$Q_{N_+}(z_0) = \left( \left[ \sum_{I=1}^{\mu} \frac{d_I}{1 - \frac{z_0}{p_I} - \frac{q_I}{p_I} \frac{z_0}{\epsilon}} \right] \left( \frac{1}{1 - \frac{z_0}{\epsilon}} \right) \right)^{N_+}. \tag{50}$$

We have used short hand notations $d_I \equiv d_I^{1,1}$ which are essentially the solutions of Eq. (18). When $q_I = 0$ for all species, it is straightforward to check that the partition function in Eq. (50) reduces to the partition function Eq. (31) of the totally asymmetric case, under the condition $\bar{\rho} = 0$.

## 5.3 Species densities, drift current and flip current

Just like we did in the $\mu$-TASEP-IAF, we can analytically calculate several observables of interest in the partially asymmetric case also, using the matrix algebra (46) and matrix representations (48) following the same procedures as before. Starting from the initial configuration stated in Eq. (49), the average density $\rho_I$ of any non-conserved species $I$ ($I = 1, \ldots, \mu$) for the $\mu$-ASEP-IAF is obtained as

$$\rho_I = \rho_+ \frac{d_I}{\left(1 - \frac{z_0}{p_I} - \frac{q_I}{p_I} \frac{z_0}{\epsilon}\right)} \frac{1}{\sum_{K=1}^{\mu} \frac{d_K}{\left(1 - \frac{z_0}{p_K} - \frac{q_K}{p_K} \frac{z_0}{\epsilon}\right)}}, \tag{51}$$

where $\rho_0$ and $\rho_+$ are the conserved densities for the vacancies and the impurities, respectively. If we put $q_I = 0$ for all $I$ in Eq. (51), the expression of $\rho_I$ for the totally asymmetric case [Eq. (35)] is correctly recovered. The drift currents $J_{I0}$ and the flip currents $J_{I \leftrightarrow K}$ for the $\mu$-ASEP-IAF ($I, K = 1, \ldots, \mu$) are given by

$$\begin{aligned}
J_{I0} &= z_0 \, \rho_+ \frac{d_I}{\left(1 - \frac{z_0}{p_I} - \frac{q_I}{p_I} \frac{z_0}{\epsilon}\right)} \frac{1}{\sum_{K=1}^{\mu} \frac{d_K}{\left(1 - \frac{z_0}{p_K} - \frac{q_K}{p_K} \frac{z_0}{\epsilon}\right)}}, \\
J_{I \leftrightarrow K} &= \rho_+ \frac{(d_I w_{IK} - d_K w_{KI})}{\sum_{K=1}^{\mu} \frac{d_K}{\left(1 - \frac{z_0}{p_K} - \frac{q_K}{p_K} \frac{z_0}{\epsilon}\right)}}.
\end{aligned} \tag{52}$$

## 5.4 Negative differential mobility

In what follows, we will show that the species in the $\mu$-ASEP-IAF under the periodic boundary condition, exhibit negative differential mobility [75,76]. More precisely, we would see that both the drift currents and the flip current can decrease with increasing bias (which we define later),

giving rise to the phenomena of negative differential mobility (NDM). NDM has been observed for driven tracer particles in the presence of static obstacles [78,79] or in crowded medium [77] and for many particle systems in presence of kinetic constraints [80] or obstacles [81]. There have been many studies to understand the mechanism of NDM in driven systems and it appears that some kind of trapping that leads to decrease in dynamical activity, acts as a main cause of NDM [79,82,83]. In connection to asymmetric simple exclusion process, a two dimensional variant of ASEP where the kinetic constraint is implemented by restricting the motion of the particles depending on the number of its occupied neighbors, has been shown to exhibit NDM at high density and high bias values [80]. In one dimension, a single driven tracer hopping asymmetrically in the environment of bath particles executing symmetric exclusion process, exhibits negative differential mobility as well as absolute negative mobility (current flowing in a direction opposite to the bias direction), where the kinetic constraint is imposed by an additional exchange dynamics of the tracer with a distant bath particle depending on the vacant nearest neighbors [84]. Another way to incorporate the effect of the kinetic constraint leading to NDM, is to consider the escape rate from a configuration as a decreasing function of the bias, shown elaborately for a biased random walker in Ref. [79].

Recently in Ref. [85], the authors have proposed that slowing down of non-driven degrees of freedom (modes) through the biasing of the driven mode, can give rise to negative differential mobility for both the driven and non-driven degrees of freedom in an interacting many particle system. Here, we apply this mechanism to show that indeed the $\mu$-ASEP-IAF can exhibit NDM for particular choices of the rates in the microscopic dynamics.

To illustrate NDM in $\mu$-ASEP-IAF, we will focus on the $\mu = 3$ case. We have three species of particles ($I = 1, 2, 3$), impurities ($+$) and vacancies in the system following the microscopic dynamics Eq. (45). We choose the drift rates $p_I$ and $q_I$ of the species $I = 1, 2, 3$ to be

$$p_1 = 1, q_1 = e^{-b}, \qquad p_2 = \frac{1}{1+b^2} = q_2, \qquad p_3 = 1 = q_3. \qquad (53)$$

The choices of the hopping rates in Eq. (53) are inspired by similar choices in Ref. [85] in context of NDM for different models. The special choices of the hopping rates in Eq. (53) allow us to identify the parameter $b$ as the hopping bias in the system. This is because $\ln(p_1/q_1) = b$ and the unbiased case $p_1 = q_1 = 1$ corresponds to $b = 0$. Then, the species 1 particle is a driven mode for any $b > 0$. Even when $b > 0$, Eq. (53) clearly states that species 2 and species 3 particles are non-driven modes in the system because the right and left hopping rates are equal for both of them. However, there is a key difference between the hopping rates of species 2 and species 3 particles. The hopping rates of species 2 depend explicitly on the bias $b$ of the driven mode (species 1). More precisely, the hopping rates $p_2$ and $q_2$ decrease with increasing bias $b$. This corresponds to the slowing down of non-driven mode and turns out to be the key for negative differential mobility. The hopping rates $p_3$ and $q_3$ of the other non-driven mode species 3, does not depend on $b$. Here we should mention the presence of another driven mode in the system, which is the impurity. Since, the impurity motion is only unidirectional, it is a driven mode by construction. Consequently, even at $b = 0$ the system is in a non-equilibrium steady state for $\epsilon > 0$ where the impurity acts as the lone driven mode. In the present context, we choose $\epsilon$ to be a constant independent of the bias $b$. We focus on the behavior of the currents with the variation of $b$. To summarize, the driven modes in the 3-ASEP-IAF are (i) species 1 (driven by $b$) and (ii) impurity (driven due to unidirectional motion with constant rate $\epsilon$, independent of $b$), whereas the non-driven modes are (i) species 2 (hopping rates are decreasing function of $b$) and (ii) species 3 (hopping rates independent of $b$). All the flip rates $w_{IK}$ ($I, K = 1, 2, 3$) are kept constants independent of the drift bias

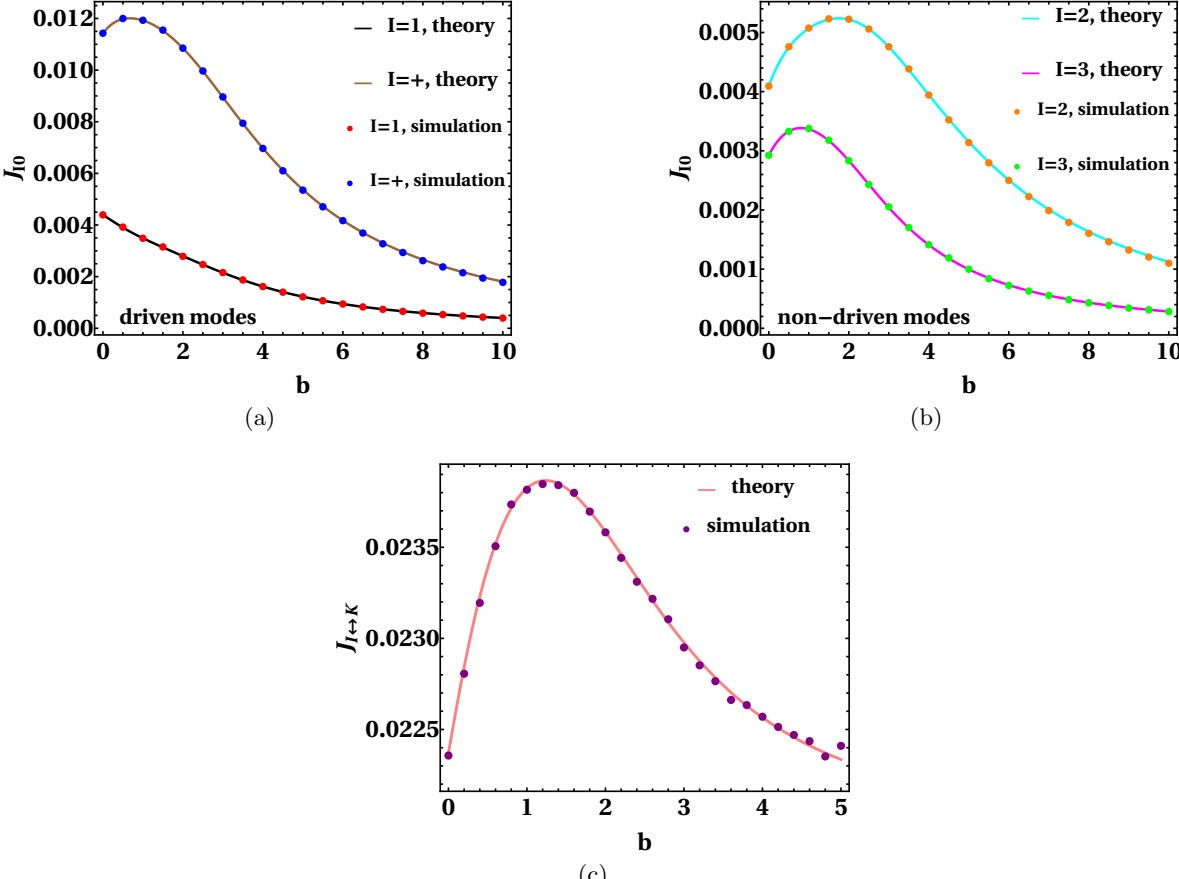

(a)                                                                                                (b)

(c)

Figure 10: The figure illustrates the negative differential mobility of the currents in the 3-ASEP-IAF. In (a), the drift current of the species 1 (driven by bias $b$) monotonically decreases with increasing bias. The current of the other driven mode (driven due to unidirectional motion with rate $\epsilon$) impurity $(+)$ shows non-monotonic behavior, it increases initially but ultimately decreases with increasing bias for large values of $b$. In (b), the drift current of both non-driven modes (species 2 and species 3) decrease with increasing bias $b$ for large values of the bias. The figure (c) shows that the flip current between any pair of species also exhibits negative differential mobility with increasing $b$. Although the flip current is not related to the drift bias $b$ directly, still it decreases with increasing bias for large values of $b$. The parameters used here are $L = 10^3, \epsilon = 0.1, p_3 = q_3 = 1.0, w_{31} = w_{12} = 0.8, w_{13} = w_{32} = 0.2, w_{21} = w_{23} = 0.5, \rho_+ = 0.3, \rho_0 = 0.4$. The ensemble average is done over $10^6$ samples.

*b*. With this set up, we now investigate the variation of the drift currents and flip current as functions of the bias *b*, both from analytical formulae Eq. (52) and Monte Carlo simulations.

In Fig. 10(a), we present the behaviors of the drift currents of the driven modes with variation of the bias *b*, under the periodic boundary condition. Interestingly, although the bias *b* is directly applied to species 1 to increase its current, the drift current for species 1 decreases monotonically with increasing bias giving rise to the phenomena of negative differential mobility. The current of the other driven mode, the impurities, initially increase with increasing bias, reaches to a maximum, but then decreases as the bias is further increased, thereby leading to NDM. The drift currents of both the non-driven modes exhibit non-monotonic behaviors with increasing bias as shown in Fig. 10(b). Both of them decrease with increasing bias for sufficiently large values of *b*, showing negative differential mobility. Notably, the flip dynamics is not directly affected by the drift bias since all the flip rates are kept to be constants independent of *b*. Therefore, it is intriguing to observe that the net flip current still decreases with increasing bias (for large *b*) and therefore exhibits NDM, as presented in Fig. 10(c). The mechanism behind the negative differential mobility in drift current is related to the decreasing dynamical activity (number of hops per unit time) of the species 2 particles (one of the non-driven modes) with increasing forward bias *b* for the species 1 particles. Since with increasing *b*, the hop rate of species 2 [Eq. (53)] decreases, its waiting time at the residing site increases i.e. it becomes more and more prone to stay at the residing site rather to leave the site as *b* increases. That is why, although the increasing bias tries to push particles forward, their ways are blocked by the slowed down species 2 particles. The exclusion interaction and the non-overtaking dynamics facilitates the NDM even better by not allowing other species or impurities to overtake the slowed down species 2 particles. The reason behind the negative differential mobility occurring in the flip current requires further investigation.

We end this section with mentioning the possibility of further nontrivial transport properties in the steady state of the $\mu$-ASEP-IAF when one considers the counter flow scenario. The counter flow in the system arises when the net bias of some species of particles are opposite to that of the others. For example, in the 2-ASEP-IAF, species 1 can have net bias to right i.e. $p_1 > q_1$ whereas the species 2 can have net bias in the opposite direction i.e. $q_2 > p_2$. Counter flow can give rise to interesting physical features e.g. phase transitions [48, 52, 86]. This urges for detailed investigation of the counter flow situation in $\mu$-ASEP-IAF in future works.

# 6 Summary and future directions

In this article, we have obtained an exact steady state probability distribution of the $\mu$-ASEP-IAF on a one dimensional lattice under periodic boundary conditions, using the matrix product ansatz. The $\mu$-ASEP-IAF consists of (i) drift of the species $(I = 1, 2, \ldots, \mu)$ and impurities, and (ii) flip between different species initiated by the impurities. In steady state, we provide the explicit finite dimensional $[(\mu + 1) \times (\mu + 1)]$ matrix representations for any $\mu > 0$ for the totally asymmetric case i.e. $\mu$-TASEP-IAF. For the partially asymmetric scenario i.e. $\mu$-ASEP-IAF, we obtain the corresponding matrices with infinite dimensional representations. Importantly, due to the non-ergodicity of the $\mu$-ASEP-IAF dynamics, the partition function and observables in the steady state depend on the specific choice of the

initial configuration. However, for a special class of initial configurations, we could indeed analytically calculate the partition function for both the totally asymmetric and partially asymmetric cases with any $\mu > 0$, under periodic boundary conditions. We present exact analytical expressions for steady state observables like the average densities of the non-conserved species, drift current, flip current and some other two-point correlations. We show that our analytical calculations are in agreement with the Monte Carlo simulations for the analytically tractable specific initial configuration. In this connection, the non-ergodicity of the model has been established extensively (Monte Carlo simulations) by showing the deviations of the steady state observable values for a random initial configuration from that of the specific initial configuration mentioned above. Along with the important exactly solvable analytical structure, the $\mu$-ASEP-IAF also has interesting physical features. Notably, with the variation of vacancy density, several two-point correlations exhibit interesting behaviors e.g. transiting between negative and positive correlations, showing both local maximum and local minimum etc. The effect of the drift on the flip processes are evident from the functional dependence of the species densities on the flip rates. Interestingly, both the drift current and flip current in the $\mu$-ASEP-IAF are shown (analytically and numerically) to display negative differential mobility (decreasing current with increasing bias) for certain choices of the drift rates. The mechanism behind the negative differential mobility relies on slowing down a non-driven mode in the system through the biasing of a driven mode, which eventually leads to decreased dynamical activity of all the modes in the steady state.

Apart from its own intriguing mathematical and physical characteristics, the $\mu$-ASEP-IAF studied here is relevant in two other important contexts. The $\mu$-ASEP-IAF has interesting connections to (i) multi lane asymmetric simple exclusion proces ($m$-ASEP) which serves as a simple yet remarkable model for multi lane traffic flow, and (ii) enzymatic chemical reactions. For the totally asymmetric model $\mu$-TASEP-IAF, these connections are discussed in details in Appendix A and Appendix C respectively. Importantly, the exact solution of $\mu$-ASEP-IAF suggests possible exact solutions in corresponding multi lane ASEP and traffic models with correlations between particles in different lanes and non-zero net current between lanes. The detailed and rigorous analysis to develop these connections and incorporate them for studying multi-lane traffic flow, constitutes one of the main future directions. We also propose a variation of the $\mu$-ASEP-IAF that exhibits better prospects for being a model for multi lane traffic flow (see Appendix B). In future we plan to investigate the exact steady state and observables of this varied $\mu$-ASEP-IAF model using matrix product ansatz. Just like the multi lane traffic flow, the connections between $\mu$-ASEP-IAF and enzymatic chemical reactions both in steady state as well as dynamics, should be analyzed in more details by considering observables relevant for the chemical reactions. In the present article, we have considered impurities with fixed finite density and constant hopping rate. It would be useful to explore the effects of the variations of impurity density and impurity hopping rate on the observables in the $\mu$-ASEP-IAF. Another important future direction would be to analyze the exact steady state of the $\mu$-ASEP-IAF with open boundary conditions which is more pertinent in context of transport processes, and might also lead to rich phase transitions. It would be interesting to look into the effect of counter flow (i.e. some species having net bias in opposite direction relative to the other species) on the transport properties and possibility of phase separations in the $\mu$-ASEP-IAF. We would also like to investigate the dynamics of the $\mu$-ASEP-IAF in detail, in particular if the product form of the steady state also prevails in the dynamics (using dynamical matrix product ansatz), the dynamical activity in terms of large deviations and possibility of dynamical phase transitions in related models.

# Acknowledgements

We thank Kazuaki Takasan for fruitful discussions. We gratefully acknowledge Arvind Ayyer for pointing out important references. This work is partially supported by the Grants-in-Aid for Scientific Research (Grant No. 21H01006).

# A    Connection between $\mu$-TASEP-IAF and multi lane TASEP

Here we explore the connections between the multi lane totally asymmetric simple exclusion process ($m$-TASEP) and the one dimensional $\mu$-ASEP-IAF. For simplicity, we consider $\mu = 2$ i.e. the 2-TASEP-IAF (and correspondingly 2-TASEP or two lane TASEP).

In Fig. 11, we present a two lane TASEP where particles can hop in forward directions in lane 1 and lane 2 with rates $p_1$ and $p_2$ respectively. The particles in lane 1 and lane 2 can be interpreted as two types i.e. species 1 and species 2 particles in the 2-TASEP-IAF process. Except for the hopping of particles in the two lanes in the 2-TASEP, the

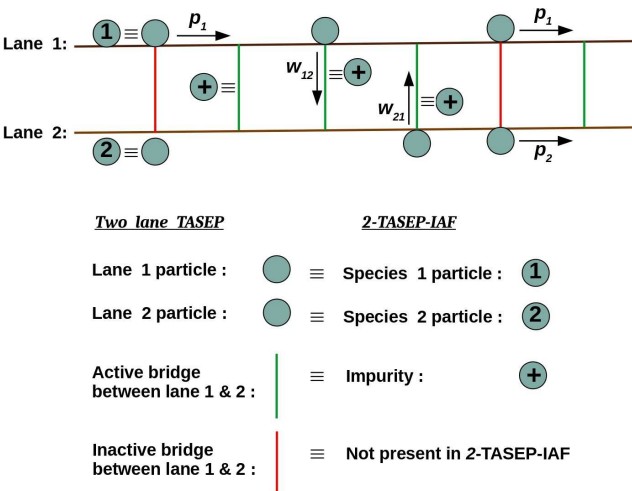

Figure 11: The figure illustrates a two lane totally asymmetric simple exclusion process and the identification of its components to the equivalent constituents of the 2-TASEP-IAF.

particles can change lanes through bridges connecting the lanes. There are two types of bridges, *active* (green vertical lines in Fig. 11) that allows vertical hopping i.e. lane change of particles and *inactive* (red vertical lines in Fig. 11) that does not allow lane change of particles. The active bridges in the two lane TASEP mimic the impurities (+) in the 2-TASEP-IAF. However, the inactive bridges are not counted in the equivalent 2-TASEP-IAF. Notably, a neighboring pair of (active, inactive) bridges can change to (inactive, active). This inactive-active transformation of neighboring lanes can be interpreted as a resultant drift of the active bridges through the system. Consequently, this accounts for the forward hopping of impurity in the 2-TASEP-IAF.

The equivalence of the microscopic dynamics of the two lane TASEP and the 2-TASEP-IAF is shown in Fig. 12. The last (bottom) panel in Fig. 12 exhibits the connection between

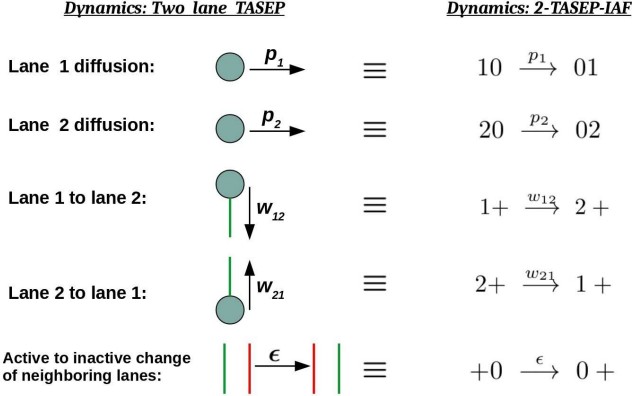

Figure 12: The figure shows the connection between each microscopic dynamics of the two lane TASEP with the equivalent microscopic dynamics in the 2-TASEP-IAF.

the inactive-active lane transformations in 2-TASEP and the impurity hopping in the 2-TASEP-IAF. The two panels above the bottom panel in Fig. 12 describe the equivalence of the lane change of particles in 2-TASEP with the impurity activated flip in the corresponding 2-TASEP-IAF. To elaborate, when the lane 1 (2) particle in 2-TASEP comes in contact with an active bridge, it can go to lane 2 (1) with rate $w_{12}$ ($w_{21}$). Similarly, when a species 1 (2) particle in the 2-TASEP-IAF encounters an impurity as a right neighbor, it can flip to a species 2 (1) particle with rate $w_{12}$ ($w_{21}$). The first two panels in Fig. 12 present the relations between usual hopping dynamics in the two processes.

We can generalize the approaches described above to establish connections between the multi-lane TASEP and the $\mu$-TASEP-IAF for any $\mu \geqslant 3$. It is noteworthy that, for the multi-species case ($\mu \geqslant 3$), we have shown the existence of non-zero net flip current in the $\mu$-TASEP-IAF. It implies the existence of net non-zero lane change current between neighboring lanes in the multi lane TASEP. Also, the correlations between different species of particles and vacancies in the $\mu$-TASEP-IAF suggest non-zero correlations between particles in the different lanes in the $m$-TASEP. The mapping can also be extended for the partially asymmetric motion of particles. We must mention that the connections between the muti-lane TASEP and $\mu$-TASEP-IAF described here, are approximate. To establish more accurate relations between the two processes, one has to perform rigorous calculations for observables in the multi-lane TASEP and compare the corresponding results with that of the $\mu$-TASEP-IAF.

## B   A variation of $\mu$-TASEP-IAF, connection to multi-lane traffic flow

The multi lane TASEP has been widely regarded as a simplistic yet important model for multi lane traffic flow [2]. Due to the connections between the $\mu$-TASEP-IAF and the multi-lane TASEP discussed in Appendix A, it is natural to ask about the applicability of $\mu$-TASEP-IAF [Eq. (1)] as a suitable model for multi lane traffic flow. Before addressing this question, we should mention that the lane change dynamics in realistic traffic flow must facilitate the traffic as a whole. More precisely, the change of lanes should increase the total flow or total current

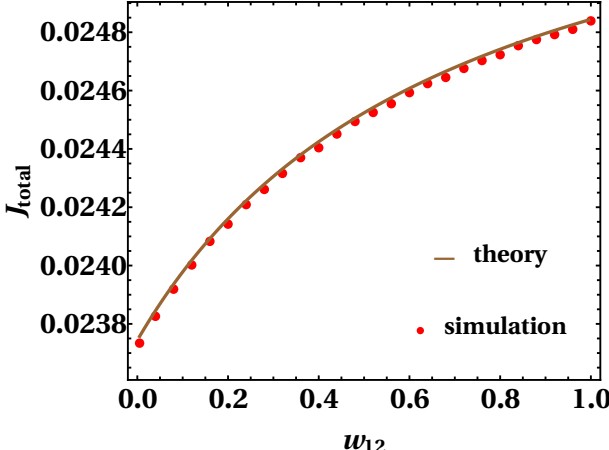

Figure 13: The figure shows the variation of the total drift current of species 1 and 2 with the flip rate $w_{12}$. While the flip rate (equivalent lane change rate in two lane TASEP) is varied over a large range $w_{12} \in (0,1)$, the corresponding increase in the total drift current (total flow along the two lanes in TASEP) is reasonably small. The parameters used are $L = 10^4, p_1 = 0.3, p_2 = 1.0, \epsilon = 0.1, w_{21} = 1.0, \rho_+ = 0.2, \rho_0 = 0.2$. The ensemble average is done over $10^7$ samples.

along the lanes. To investigate this for the $\mu$-TASEP-IAF with $\mu = 2$, we plot the total drift current of species 1 and species 2, $J_{\text{total}} = J_{10} + J_{20}$ as a function of the flip rate $w_{12}$ in Fig. 13. Of course, for the two lane TASEP, this amounts to investigating the variation of the total drift current of lane 1 and lane 2 by changing the lane change rate $w_{12}$.

In Fig. 13 we observe that the total current, although increases with the flip rate, the amplitude of the increment is quite small keeping in mind the wide range of variation in the tuning parameter $w_{12} \in (0,1)$. The reason behind this, as revealed by a careful observation, is the *approximate* mapping between the $\mu$-TASEP-IAF and multi lane TASEP described in Fig. 12. In the $\mu$-TASEP-IAF dynamics, when a species encounters an impurity, it flips but does not change its position. On the other hand, in the multi lane TASEP, when a particle in any lane comes in contact with an active bridge, it actually changes the lane i.e. not only changes its characteristics (lane 1 particle to lane 2 particle or vice versa) but also changes its position. To incorporate this in our present model, we propose a variation of the $\mu$-TASEP-IAF dynamics in Eq. (1). Specifically, the change is made only in the flip dynamics. Earlier in Eq. (1), the flip dynamics has been

$$I+ \quad \overset{w_{IK}}{\longrightarrow} \quad K+, \tag{54}$$

with $I, K = 1, ..., \mu$ and $I \neq K$. Whereas we propose the new flip dynamics to be

$$I+ \quad \overset{w_{IK}}{\longrightarrow} \quad +K. \tag{55}$$

Note that, in comparison to Eq. (54), the flip process in Eq. (55) accompanies the flip of the species with a hop towards right. The other hopping processes in Eq. (1) remain the same for this varied $\mu$-TASEP-IAF. The effect of the dynamics can be immediately observed in Fig. 14 where we present the variation of the total drift current as a function of the flip rate (lane change rate) for both the dynamics in Eq. (54) and Eq. (55) (using Monte Carlo simulations).

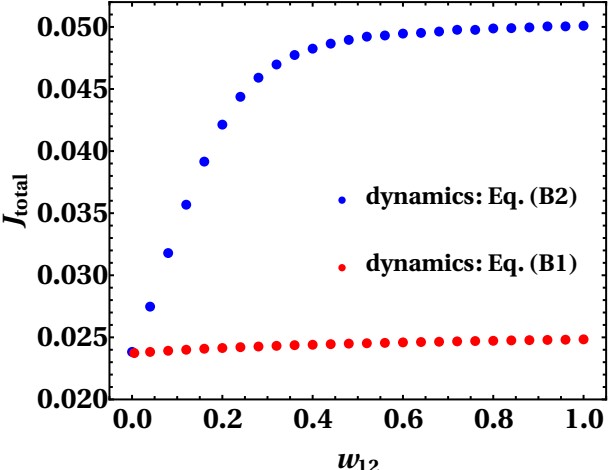

Figure 14: The figure illustrates the comparison of the total drift current $J_{\text{total}}$ between the flip process in Eq. (54) and the proposed variation in Eq. (55). The current $J_{\text{total}}$ for the variation Eq. (55) increases considerably with increasing flip rate (equivalent lane change rate in two lane TASEP) $w_{12}$, whereas the same for the original flip process Eq. (54) increase much slowly (Fig. 13). The parameters used are $L = 10^4, p_1 = 0.3, p_2 = 1.0, \epsilon = 0.1, w_{21} = 1.0, \rho_+ = 0.2, \rho_0 = 0.2$. The ensemble average is done over $10^7$ samples.

Indeed, the Fig. 14 shows that $J_{\text{total}}$ increases considerably with increasing lane change rate for Eq. (55) whereas it grows weakly for Eq. (54) (see Fig. 13). This observation implies that the proposed variation of the $\mu$-TASEP-IAF acts as a better model for multi lane traffic flow in comparison to the original model. It would be interesting to study this variation of the $\mu$-TASEP-IAF both analytically and numerically and to build connections with the multi lane traffic flow.

## C    Connection between $\mu$-TASEP-IAF and enzymatic chemical reactions

In this appendix, we briefly discuss some connections between the $\mu$-TASEP-IAF and enzymatic chemical reactions. One of the simplest form of the enzymatic chemical reaction is,

$$E + S \underset{k_b}{\overset{k_f}{\rightleftharpoons}} ES \underset{\bar{k}_b}{\overset{\bar{k}_f}{\rightleftharpoons}} E + P, \tag{56}$$

where $E, S, P$ denotes enzyme, substrate, product respectively and $ES$ corresponds to the intermediate complex. The parameters $k_f$ and $k_b$ are the rate constants for forward and backward reaction for the intermediate complex formation, while $\bar{k}_f$ and $\bar{k}_b$ are the rate constants for forward and backward reactions between the intermediate complex and the product (along with enzyme). Clearly, the initially present enzyme in the reaction remains intact after the reaction is completed. Here, to discuss some connections to the $\mu$-ASEP-IAF,

Figure 15: The figure illustrates the connection between the simplified form of the enzymatic chemical reaction Eq. (57) (in a narrow channel with many units of drifting enzymes, substrates and products) and the 2-TASEP-IAF. The enzyme, substrate and product in the chemical system can be identified as impurity, species 1 and species 2 particle in the 2-TASEP-IAF.

we would rather consider a much simplified version of the chemical reaction (56) as

$$S + E \underset{k_P}{\overset{k_S}{\rightleftharpoons}} P + E, \tag{57}$$

where we ignore the intermediate complex formation. $k_S$ $(k_P)$ is the rate constant for $S$ transforming to $P$ ($P$ transforming to $S$).

Let us consider a spatially extended narrow channel with many units of substrates, enzymes and products all of which drift through the channel at different rates. This system of chemical reagents can be approximately mapped to an equivalent $\mu$-TASEP-IAF. As explained in Fig. 15 for $\mu = 2$, the impurity in 2-ASEP-IAF plays the role of the enzyme, as it can transform one species of particle to another species. One species, e.g. species 1 can be considered as $S$ for chemical reaction Eq. (57), whereas the species 2 particle acts as $P$. The flip rates $w_{12}$ and $w_{21}$ mimic the rate constants $k_S$ and $k_P$. With this set up, our study of the 2-TASEP-IAF reveals the effect of drift on the resultant concentrations of substrates and products in the steady state. Notably, the multi-species ($\mu > 2$) case of the $\mu$-TASEP-IAF can be mapped to generalized version of the chemical reaction Eq. (57) as

$$S_i + E \underset{k_{P_j}}{\overset{k_{S_j}}{\rightleftharpoons}} P_j + E, \tag{58}$$

where $i = 1, \ldots, \mu_1$ are identified as $\mu_1$ number of substrates and $j = 1, \ldots, \mu_2$ are identified as $\mu_2$ number of products in the enzymatic chemical reaction system (with $E$ acting as the enzyme for each chemical reaction) where $\mu_1 + \mu_2 = \mu$ ($\mu$ being the total number of species in the $\mu$-TASEP-IAF). To study the time evolution of the enzymatic chemical reaction, one has to study the dynamics of the $\mu$-TASEP-IAF. The connection between the $\mu$-TASEP-IAF and the enzymatic chemical reactions should be studied more thoroughly with proper attention to the observables of interest for the chemical reactions.

# D  Solution of Eq. (26) for fugacity $z_0$: special cases

We have calculated the partition function (Sec. 3) and observables (Sec. 4) in the grand canonical ensemble, by associating a fugacity $z_0$ with the vacancies. Since the observables have to be finally expressed in terms of the input parameters $(p_1, p_2, \epsilon, w_{12}, w_{21}, \rho_0, \rho_+)$ only, an important step in the calculation is to solve Eq. (26) to obtain the fugacity as a function of these input parameters i.e. $z_0(p_1, p_2, \epsilon, w_{12}, w_{21}, \rho_0, \rho_+)$. However, in most cases the solutions of $z_0$ from Eq. (26), cannot be obtained explicitly. In this appendix, we would provide two simple cases for particular choices of the hop-rates and the initial configuration where the solutions for $z_0$ get simplified significantly. Specifically, we would consider $\bar{\rho} = 0$ for the initial configuration (see Sec. 3). Consequently, the density-fugacity relation Eq. (26) becomes a *quartic* equation in the variable $z_0$, emerging from

$$\rho_0 = \rho_+ z_0 \left[ \frac{1}{\epsilon - z_0} + \frac{1}{p_1 w_{21}(p_2 - z_0) + p_2 w_{12}(p_1 - z_0)} \left( p_1 w_{21}\frac{p_2 - z_0}{p_1 - z_0} + p_2 w_{12}\frac{p_1 - z_0}{p_2 - z_0} \right) \right]. \tag{59}$$

Below we discuss two special cases.

## D.1  Case I:

A particularly simple solution can be acquired for the choice $p_1 = p_2 = 1 \neq \epsilon$. As a result, Eq. (59) is reduced to a *quadratic* equation which leads to the following solution

$$z_0 = \frac{(1 + \epsilon)(\rho_0 + \rho_+) - \sqrt{(1 + \epsilon)^2(\rho_0 + \rho_+)^2 - 4\epsilon\rho_0(\rho_0 + 2\rho_+)}}{2(\rho_0 + 2\rho_+)}. \tag{60}$$

Note that in this case, the fugacity does not depend explicitly on the flip rates $w_{12}$ and $w_{21}$.

## D.2  Case II:

A comparatively cumbersome yet closed form solution is attained for the case $p_1 = 1/2, p_2 = \epsilon = w_{21} = 1, \rho_+ = 1/4$. Here the fugacity would be a function of $w_{12}$ and $\rho_0$ i.e. $z_0(w_{12}, \rho_0)$. The reason behind keeping $w_{12}$ and $\rho_0$ as free parameters is that, the observables in the main text have been mostly analyzed as functions of these two parameters. The corresponding solution (Eq. (59) essentially reduces to a cubic equation) for the fugacity is given below,

$$z_0 = -\frac{a_2}{3a_3} + \frac{2^{1/3}(-a_2^2 + 3a_3 a_1)}{3a_3|\nu|} - \frac{2^{2/3}|\nu|}{6a_3}, \tag{61}$$

where $|\nu|$ denotes the absolute value of $\nu$ and its functional form is

$$\nu(a_0, a_1, a_2, a_3) = \\ (-2a_2^3 + 9a_3 a_2 a_1 - 27a_3^2 a0 + \sqrt{-4(a_2^2 - 3a_3 a_1)^3 + (2a_2^3 - 9a_3 a_2 a_1 + 27a_3^2 a_0)^2})^{\frac{1}{3}}. \tag{62}$$

In Eqs. (61) and (62), the parameters $a_0, a_1, a_2, a_3$ are explicit functions of $w_{12}$ and $\rho_0$, as follows

$$\begin{aligned} a_0 &= -4\rho_0(1 + w_{12}), \\ a_1 &= 3 + 2w_{12} + 4\rho_0(4 + 5w_{12}), \end{aligned}$$

$$
\begin{aligned}
a_2 &= -7 - 8w_{12} - 4\rho_0(5 + 8w_{12}), \\
a_3 &= 4(1 + 2\rho_0)(1 + 2w_{12}).
\end{aligned}
\tag{63}
$$

Although there seems to be no definite rule for obtaining closed-form solutions of $z_0$ like Eqs. (60) and (61), one might achieve other convenient solutions by searching for suitable subspace of the transition rates.

# E Block-diagonal structure of the transition rate matrix

In this appendix, we show the block-diagonal structure of the transition rate matrix $M$ [Eq. (3)], reflecting the non-ergodicity of $\mu$-ASEP-IAF. To illustrate this with an example for $\mu = 2$, we consider a small system of size $L = 4$ where the number of impurity and vacancy are given by $N_+ = 1$ and $N_0 = 1$, respectively, and the total number of species 1 and species 2 particles is $N_1 + N_2 = 2$. Total number of configurations in the configuration space, in this case, is 48. However, since there is no spatial disorder in the transition rates, we take into account the translational invariance of the model on a periodic lattice. Consequently, there are 12 independent configurations of the system, which we denote as follows (depending on the sequence of species 1 and 2)

$$
\begin{aligned}
11 + 0 &\equiv I_1, & 110+ &\equiv I_2, & 101+ &\equiv I_3, \\
12 + 0 &\equiv II_1, & 120+ &\equiv II_2, & 102+ &\equiv II_3, \\
21 + 0 &\equiv III_1, & 210+ &\equiv III_2, & 201+ &\equiv III_3, \\
22 + 0 &\equiv IV_1, & 220+ &\equiv IV_2, & 202+ &\equiv IV_3.
\end{aligned}
\tag{64}
$$

We have divided the 12 configurations in Eq. (64) into 4 sectors $I, II, III, IV$ where the three configurations within a given sector are connected through the *drift dynamics*. To investigate the connectivity between these sectors through the *flip dynamics*, below we provide the full transition rate matrix for these 12 configurations (enumerated consecutively from $I_1$ to $IV_3$),

$$
M = \begin{pmatrix} M_{I,II} & \bigcirc \\ \bigcirc & M_{III,IV} \end{pmatrix},
\tag{65}
$$

where

$$
M_{I,II} = \begin{pmatrix}
-\epsilon - w_{12} & 0 & p_1 & w_{21} & 0 & 0 \\
\epsilon & -p_1 & 0 & 0 & 0 & 0 \\
0 & p_1 & -p_1 - w_{12} & 0 & 0 & w_{21} \\
w_{12} & 0 & 0 & -\epsilon - w_{21} & 0 & p_1 \\
0 & 0 & 0 & \epsilon & -p_2 & 0 \\
0 & 0 & w_{12} & 0 & p_2 & -p_1 - w_{21}
\end{pmatrix},
$$

$$
M_{III,IV} = \begin{pmatrix}
-\epsilon - w_{12} & 0 & p_2 & w_{21} & 0 & 0 \\
\epsilon & -p_1 & 0 & 0 & 0 & 0 \\
0 & p_1 & -p_2 - w_{12} & 0 & 0 & w_{21} \\
w_{12} & 0 & 0 & -\epsilon - w_{21} & 0 & p_2 \\
0 & 0 & 0 & \epsilon & -p_2 & 0 \\
0 & 0 & w_{12} & 0 & p_2 & -p_2 - w_{21}
\end{pmatrix},
\tag{66}
$$

and $\bigcirc$ is $6 \times 6$ null matrix. In Eqs. (65) and (66), we clearly observe that the transition rate matrix in in block-diagonal form with *two* blocks. We observe that sector $I$ is connected to

sector $II$ through flip dynamics, whereas sector $III$ and $IV$ are also connected to each other via flip dynamics. However, sectors $(I, II)$ are disconnected from sectors $(III, IV)$, thereby creating two separate blocks in the rate matrix.

Note that, in absence of the flip dynamics (i.e. $w_{12} = w_{21} = 0$), sectors $I$ become disconnected from $II$, similarly $III$ gets disconnected from $IV$, resulting in four blocks in the transition matrix. On the other hand, in the special limit when $N_1 + N_2 = 1$, we would have a single block with the system becoming ergodic.

Next we explore the variation in the number of blocks as the system size is increased. We keep $N_0 = 1$ throughout, because it appears that the number of blocks depends on the arrangements of $1, 2$ and $+$, but not on the location of vacancies. This might be better understood in a box-particle representation of the model where $1, 2, +$ denote boxes and 0-s are particles.

For $L = 5$, the special case $N_1 + N_2 = 1$ ($N_+ = 3$) keeps the system ergodic with a single block only. But, as we increase $N_1 + N_2$, e.g. $N_+ = 2$ and $N_1 + N_2 = 2$, one can check that the rate matrix is block-diagonal with 3 blocks. With further increase in $N_1 + N_2 (=3)$ which also corresponds to $N_+ = 1$, we have 4 blocks in the transition rate matrix. Below we present $N_{blocks}$ in a tabular form, explicitly for a few sets of $(L, N_+)$, with $N_0 = 1$ and $N_1 + N_2 = L - N_0 - N_+$,

| $L$ | $N_+$ | $N_{blocks}$ |
|---|---|---|
| 4 | 1 | 2 |
| 4 | 2 | 1 |
| 5 | 1 | 4 |
| 5 | 2 | 3 |
| 5 | 3 | 1 |
| 6 | 1 | 8 |
| 6 | 2 | 6 |
| 6 | 3 | 3 |
| 6 | 4 | 1 |
| 7 | 1 | 16 |
| 7 | 2 | 15 |
| 8 | 1 | 32 |
| 8 | 2 | 32 |
| 9 | 1 | 64 |
| 9 | 2 | 74 |
| 10 | 1 | 128 |
| 10 | 2 | 160 |

In fact, for fixed system size $L$, with $N_0 = 1$, the general formulae for number of blocks $N_{blocks}$ in the transition rate matrix, for cases $N_+ = 1$ and $N_+ = 2$ turn out to be

$$
\begin{aligned}
N_+ = 1 : \qquad & N_{blocks} = 2^{L-3}, \\
N_+ = 2 : \qquad & N_{blocks} = 2^{L-6} L, \quad L \ \text{ even} \\
& \qquad\quad = 2^{L-6}(L-1) + 2^{\frac{L-7}{2}}\left(2^{\frac{L-5}{2}} + 1\right), \quad L \ \text{ odd}, \\
N_+ = L - 2 : \qquad & N_{blocks} = 1.
\end{aligned}
\tag{67}
$$

It would be interesting to find out the analytical formula for the number of blocks in the transition rate matrix for any general $N_+$, which would contain the formulae in Eq. (67) as special cases.

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
