# Peer review of "Multi species asymmetric simple exclusion process with impurity activated flips"

_SciPost Physics_

## Round 2 · Referee Report · Anonymous (Referee 1) · 2022-7-8

Strengths

1- Defines a new stochastic particle model with an exact matrix product stationary state 2-Calculations of physical observables carried out. 3- Negative differential mobility elucidated

I would say the paper satisfies SciPost expectation 3.

Weaknesses

1- Perhaps a technical advance rather than a discovery of new physics

Report

The paper considers a multi-species variant of the well-studied totally asymmetric exclusion process, on a periodic lattice. The model contains "defect" particles which activate switches of particle species when particles become adjacent to them. The key finding is that the model admits a matrix product solution of the stationary state. This adds to the range of models that are known to have matrix product solutions.

Although the model is of intrinsic interest, due to its matrix product stationary state, various further motivations for the model are given in the introduction. I find the mapping to a multi-lane TASEP of most interest.

The proof of the matrix product solution follows the usual construction outlined in equations (4)-(6). The auxiliary `tilde' matrices turn out to be scalars which is a crucial simplification.

It is noted that the model is non-ergodic, meaning that the configuration space breaks up into different sectors corresponding to different initial conditions and for each sector of initial conditions the partition function has to be computed separately. The partition function is computed for a special class of initial conditions and observables such as density profiles and currents are computed using the grand canonical ensemble in the large system limit.

I find the results interesting and worthy of eventual publication in SciPost. First, the authors should consider the following points.

1) The first matrix product solution for a two species ASEP was given in Derrida, B., Janowsky, S.A., Lebowitz, J.L. , Speer E.R.. Exact solution of the totally asymmetric simple exclusion process: Shock profiles. J Stat Phys 73, 813–842 (1993). https://doi.org/10.1007/BF01052811. (I will refer to this paper as [DJLS] below.)

The work [DJLS] should certainly be cited. It also showed how the stationary state factorises about the defect (a second-class particle in that case) which implies a projector form for the matrix $A$, the same as in equations (10) and (13) of the present work. This factorisation property due to the projector form of $A$ should be acknowledged in the current paper.

2) I am not sure I understand the last sentence of Section 2 `However, we should mention that.. this is not the general expression for $\alpha$ ..'. Does this mean that generally $\alpha$ would appear as $\alpha_I$ in equation (15)? Perhaps this point can be clarified.

3) Second sentence of Section 3. I would put transfer matrix in inverted commas as this is not the same as a usual equilibrium transfer matrix. i.e. Here the "transfer matrix'' $T$ refers to ..

4) In section 3 the grand canonical ensemble is used , which results in a fugacity $z_0$ which is fixed by the density. It would be helpful to have the solution for $z_0$ appear somewhere, perhaps in an appendix if it is really very complicated. Does the solution for $z_0$ simplify in some limits?

5) Section 5.4 considers Negative Differential Mobility (NDM). Some references and discussion of specific, related models, exhibiting NDM would be appropriate e.g.

Cividini J, Mukamel D and Posch H A ``Driven tracer with absolute negative mobility'' (2018) J. Phys. A: Math. Theor. 51 085001

6) Some typos: p.3 paragraph 2 "with variety" $\to$ "with a variety"

p.4. paragraph 2 "For specific choice of" $\to$ "For a specific choice of"

p.5 last line "itlaics" $\to$ "italics"

p.7 after equation (10) "resembles to that of the defect or second class" $\to$ "resembles that of the defect of second class". Here reference [DJLS] (see point 1. above) should be cited.

p.36 reference [59] author is J. Szavits-Nossan

Requested changes

-List of 6 suggested points for improvement in the above report

Attachment

  • validity: top
  • significance: good
  • originality: good
  • clarity: high
  • formatting: good
  • grammar: good

Author:  Amit Chatterjee  on 2022-10-19  [id 2933]

(in reply to Report 1 on 2022-07-08)

We hereby attach "Response_first_referee.pdf" containing answers to the comments and suggestions by the referee.

Attachment:

Response_first_referee.pdf

---

## Round 2 · Referee Report · Anonymous (Referee 2) · 2022-9-26

Report

The authors consider a particular one dimensional multi species asymmetric simple exclusion process, which contains one species of particles ("impurities") that induces conversions of the other particle species when the latter are next to them. The main result of the work is to construct the steady state for certain initial conditions in terms of a matrix product solution. This is a new and interesting analytic result that generalizes known matrix product solutions of many-particle stochastic processes. In addition, the authors carefully compare their analytic results to Monte-Carlo simulations for a number of physical quantities and find excellent agreement. In my view these results warrant publication in SciPost Physics.

I have a couple of comments the authors may or may not want to consider.

  1. Finite dimensional MPS representations for steady states of stochastic processes were to the best of my knowledge first considered in

F.H.L. Essler and V. Rittenberg 1996 J. Phys. A: Math. Gen. 29 3375 K Mallick and S Sandow 1997 J. Phys. A: Math. Gen. 30 4513

  1. As the authors explain very clearly, the process is not ergodic and certain "motifs" are conserved under the dynamics. Does this imply that the Liouvillian is block-diagonal with respect to some underlying symmetry, and the number of blocks scales exponentially with the system size? If this is the case the situation would be somewhat similar to what has been recently observed in certain stochastic processes (both classical and quantum)

K. Klobas et al J. Stat. Mech. (2018) 123202 F.H.L. Essler and L. Piroli Phys. Rev. E 102, 062210 (2020).

If there is indeed such a symmetry structure I think it would be very nice to construct it explicitly.

Requested changes

None.

  • validity: top
  • significance: high
  • originality: high
  • clarity: high
  • formatting: good
  • grammar: good

Author:  Amit Chatterjee  on 2022-10-19  [id 2934]

(in reply to Report 2 on 2022-09-26)

We hereby attach "Response_second_referee.pdf" to answer the comments and suggestions by the second referee.

Attachment:

Response_second_referee.pdf

---

## Editorial Decision

resubmitted